# Accuracy prompts are a replicable and generalizable approach for reducing the spread of misinformation

Gordon Pennycook [1,2✉] & David G. Rand [3✉]

Interventions that shift users attention toward the concept of accuracy represent a promising approach for reducing misinformation sharing online. We assess the replicability and generalizability of this accuracy prompt effect by meta-analyzing 20 experiments (with a total $N = 26,863$) completed by our group between 2017 and 2020. This internal meta-analysis includes all relevant studies regardless of outcome and uses identical analyses across all studies. Overall, accuracy prompts increased the quality of news that people share (sharing discernment) relative to control, primarily by reducing sharing intentions for false headlines by 10% relative to control in these studies. The magnitude of the effect did not significantly differ by content of headlines (politics compared with COVID-19 related news) and did not significantly decay over successive trials. The effect was not robustly moderated by gender, race, political ideology, education, or value explicitly placed on accuracy, but was significantly larger for older, more reflective, and more attentive participants. This internal meta-analysis demonstrates the replicability and generalizability of the accuracy prompt effect on sharing discernment.

[1] Hill/Levene Schools of Business, University of Regina, Regina, SK, Canada. [2] Department of Psychology, University of Regina, Regina, SK, Canada. [3] Sloan School of Management, Massachusetts Institute of Technology, Cambridge, MA, USA. ✉email: gordon.pennycook@uregina.ca; drand@mit.edu

Online misinformation has become a major focus of attention in recent years among academics, technology companies, and policy makers. Starting with "fake news" during the 2016 U.S. Presidential Election[1], and redoubling during the COVID-19 pandemic[2–4] and the aftermath of the 2020 U.S. Presidential election[5–7], there has been widespread concern about the circulation of false and misleading content on social media. There is considerable debate about the scope and impact of the misinformation problem on social media[2,8–16] (arising in part due to different definitions of "fake news"[17]). Be that as it may, a sizable body of research has been devoted to identifying and evaluating approaches for combatting the spread of misinformation online (for reviews, see Refs. [14,18,19])

One recently proposed approach, which we focus on here, involves shifting users' attention to accuracy in an effort to improve the quality of the news they share. Reducing the sharing of misinformation is of substantial importance, because simply being exposed to misinformation can increase subsequent belief[13,20–22]. The massive networked character of social media platforms means that when people choose to share misinformation online, it has the potential to reach (and influence) a large number of others. As a result, reducing the sharing likelihood of misinformation can substantially reduce its reach (as show, for example, by agent-based simulations of network spreading dynamics[23,24]).

Recent work has observed that there is a disconnect between perceptions of accuracy and sharing intentions: Even when participants are quite good at distinguishing between true and false headlines (if they are asked to judge accuracy), this ability to discern truth from falsehood often does not translate to sharing[24–26]. Ironically, this occurs despite the fact that an overwhelming majority of participants say that it is important to only share accurate news[24]. Although factors such as animosity toward political opponents[27] and personality factors such as a "need for chaos"[28] also contribute to misinformation sharing, evidence suggests that mere inattention to accuracy plays a role in the apparent disconnect between accuracy judgments and sharing[24].

Evidence for the role of inattention comes from experiments in which prompting participants to think about the concept of accuracy – for example, by asking them to evaluate the accuracy of a random headline at the beginning of the study – reduces the disconnect between accuracy judgments and sharing intentions, and thereby increases the quality of news shared[24–26]. This effect has been replicated in pre-registered studies conducted by other research groups[29], a variety of successful accuracy prompts have been identified[25], and the effectiveness of this approach has also been demonstrated in a large field experiment on Twitter where accuracy prompts were sent to users who had been sharing low-quality news content[24]. However, questions have been raised about whether it operates by decreasing sharing of false news or increasing sharing of true news[29], whether it is moderated by individual differences relating to political ideology[24,29–32] and attentiveness[29], and whether it quickly dissipates[29].

A more systematic investigation of the accuracy prompt effect is therefore necessary. This is an important issue because accuracy prompts offer a promising approach for improving the quality of news that people share online. One appealing feature of accuracy prompts is that they are proactive, rather that reactive – by shifting users' attention towards accuracy, they can prevent the sharing of misinformation in the first place (rather than playing catch-up afterward with warnings or corrections). Another advantage is that accuracy prompts do not require stakeholders such as technology companies or government regulators to decide what is true versus false. Instead, they leverage users' own abilities to identify misinformation and take advantage of the fact that most people want to share accurate content[24].

Here, we use an internal meta-analysis to assess the replicability of the positive effects of accuracy prompts identified in previous research, as well as the generalizability[33] of these effects across accuracy prompt implementations (see Table 1), headline sets and news topics (politics versus COVID-19), subject pools (convenience samples on Amazon Mechanical Turk versus more nationally representative samples from Lucid and YouGov), and user characteristics. In addition to its theoretical importance for understanding the psychology of misinformation[18], assessing replicability and generalizability is of key practical importance: evaluating the "evidence readiness"[34] of the accuracy prompt intervention approach is essential given that technology companies are considering adopting such interventions[35], and governments and civil society organizations may do the same. Before policy makers can be encouraged to implement accuracy prompts, they must know whether the effects are replicable and generalizable.

To that end, we perform an exhaustive meta-analysis of accuracy prompt experiments that our group has conducted. There are two main threats to the validity of meta-analytic results: the systematic omission of studies (e.g., publication bias suppressing studies with null results[36,37]), and the flexible selection of analysis approaches within each study inflating the rate of false positives (e.g., p-hacking[38]). Our meta-analysis addresses both of these issues because we have complete access to all relevant data. This allows us to avoid publication bias by including all qualifying studies, regardless of their results, and avoid inflating false positives through flexible analysis by applying the exact same analytic approach for all studies (an approach that was common across preregistrations for the subset of studies that had pre-registered analysis plans). Although it has been observed that biases caused by flexibility in analyses or selection criteria may be exacerbated in internal meta-analyses[39], this is not a concern in the present case. First, there is no bias from analysis flexibility as we use the

**Table 1 Description of the various accuracy prompts used across experiments. Example materials can be found on OSF.**

| Accuracy prompt | Description |
|---|---|
| Evaluation | Participants are asked to rate the accuracy of a neutral (non-political, non-COVID-19) headline. In some variants, they are shown ten headlines instead of 1; in other variants, they are given feedback on whether their answer was correct. When subsetting analyses on the Evaluation treatment, we only include studies where a single headline was shown without feedback. |
| Importance | Participants are asked how important it is to them to only share accurate news or to not share inaccurate news. |
| Norms | Participants are told that most other survey respondents think it is very important to only share accurate news. |
| PSA video | Participants are shown a 30 s video (in the format of a "Public Service Announcement", although these words are not explicitly mentioned) reminding them to think about accuracy before sharing. |
| Reason | Participants are asked how important it is to them to only share news that they have thought about in a reasoned, rather than emotional, way. |
| Tips | Participants are shown a set of minimal digital literacy tips; for sample tips, see Ref. [25]. |

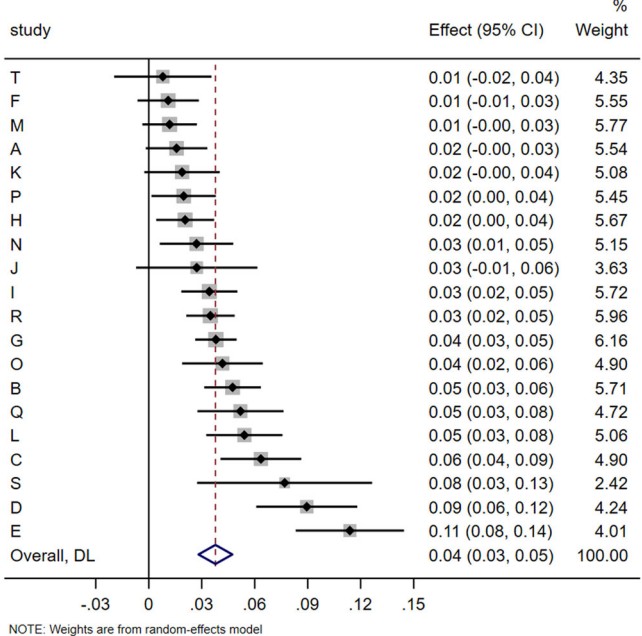

**Fig. 1 Accuracy prompts significantly increase sharing discernment.**
Meta-analytic estimate (via random effects meta-analysis) of the effect of accuracy prompts on sharing discernment across the 20 experiments analyzed in this paper. The coefficient on the interaction between condition and headline veracity and 95% confidence interval are shown for each study, and the meta-analytic estimate is shown with the red dotted line and blue diamond (positive values indicate that the treatment increased sharing discernment). We find significant heterogeneity in effect size across studies, Cochran's Q test, $Q(19) = 88.53$, $p < 0.001$, $I^2 = 78.5\%$ ($k = 20$ independent studies).

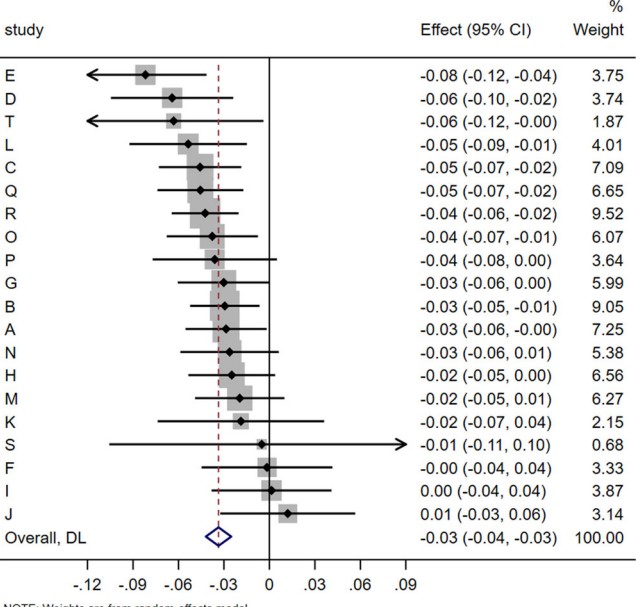

**Fig. 2 Accuracy prompts significantly decrease sharing intentions for false news.** Meta-analytic estimate (via random effects meta-analysis) of the effect of accuracy prompts on sharing of false news across the 20 experiments analyzed in this paper. The coefficient on the condition dummy (which captures the effect of the treatment on sharing of false headlines) and 95% confidence interval are shown for each study, and the meta-analytic estimate is shown with the red dotted line and blue diamond. We find no significant heterogeneity in effect size across studies, Cochran's Q test, $Q(19) = 23.33$, $p = 0.223$, $I^2 = 18.5\%$ ($k = 20$ independent studies).

In total, we meta-analyze $k = 20$ experiments, with a total of $N = 26,863$ participants (see Table 2). For details of study selection criteria and measures, see Methods. Each study in the meta-analysis had ethical approval from the Yale University IRB, the Massachusetts Institute of Technology COUHES and/or the University of Regina REB and participants provided informed consent.

## Results
We begin by examining the overall effect of the various accuracy prompts on sharing intentions across all experiments. Recall that in each study, participants were randomized to receive or not receive an accuracy prompt prior to indicating their sharing intentions for a series of true and false headlines. For analyses, sharing intentions (which were typically collected using a 6 point Likert scale) are scaled such that 0 corresponds to the minimum scale value and 1 corresponds to the maximum scale value. Thus, it is the [0,1] interval that is relevant for interpreting the magnitude of regression coefficients (e.g., if sharing intentions were binary, coefficients would be in units of percentage points). We also provide percentage changes relative to control to help contextualize the effect sizes. All statistical tests are two-tailed.

We find that accuracy prompts significantly increase sharing discernment (interaction between headline veracity and treatment dummies: $b = 0.038$, $z = 7.102$, $p < 0.001$; Fig. 1), which translates into a 71.7% increase over the meta-analytic estimate of baseline sharing discernment in the control condition (headline veracity dummy: $b = 0.053$, $z = 6.636$, $p < 0.001$). This increase in discernment was driven by accuracy prompts significantly decreasing sharing intentions for false news (treatment dummy: $b = -0.034$, $z = 7.851$, $p < 0.001$; Fig. 2), which translates into a

same analysis as was preregistered in the very first experiment for the full collection of studies. Second, there is no bias from study selection as we determined which studies to include (and when to stop including studies) simply by setting a date range (2017–2020), and did so prior to conducting the meta-analysis. Furthermore, we included all interventions that could be construed as accuracy prompts – i.e., interventions that occurred prior to the sharing task, that invoked accuracy in some way (such that the concept of accuracy would be primed), and did not provide any specific information about the veracity of any particular headlines (were "content neutral"). Importantly, although our meta-analysis was not itself pre-registered, we made the decision about what to include before conducting the meta-analysis. As a result, study selection was broad and not susceptible to motivated choices about inclusion. Our meta-analysis therefore provides an unbiased assessment of the replicability and generalizability of the impact of accuracy prompts on sharing intentions.

For this analysis, we focus largely on news sharing discernment; i.e., the extent to which the interventions improve the overall quality of news that people share, which is calculated by taking the difference between sharing intentions for true news and false news (with a higher value indicating more relative sharing of true news). This approach is superior to simply focusing on the sharing of false news because an intervention that decreases the sharing of both true and false news equally would not indicate that people are focusing more on accuracy[40]. Rather, it would indicate that people are simply more skeptical or unwilling to share any news.

**Table 2 Description of the 20 experiments included in the meta-analysis.**

| Study | Date | Sample | Platform | Topic | Accuracy prompts used | Published? |
|---|---|---|---|---|---|---|
| A* | September-17 | 847 | MTurk | Politics | -Evaluation | Unpublished |
| B* | October-17 | 1158 | MTurk | Politics | -Evaluation | Pennycook et al. 2021 S3[24] |
| C* | November-17 | 1248 | MTurk | Politics | -Evaluation | Pennycook et al. 2021 S4[24] |
| D | March-19 | 1007 | MTurk | Politics | -Importance<br>-Norms<br>-Reason<br>-Importance+Norms+Reason | Unpublished |
| E | March-19 | 1210 | MTurk | Politics | -Evaluation (10x)<br>-Evaluation (10x)<br>+Importance+Norms+Reason<br>-Evaluation (10x) with feedback<br>-Evaluation (10x) with feedback<br>+Importance+Norms+Reason | Unpublished |
| F | April-19 | 1184 | Lucid | Politics | -Evaluation (10x) with feedback<br>+Importance+Norms+Reason<br>-Evaluation+Norms<br>-Importance+Norms | Unpublished |
| G* | May-19 | 1286 | Lucid | Politics | -Evaluation<br>-Importance | Pennycook et al. 2021 S5[24] |
| H | September-19 | 2296 | MTurk | Politics | -Evaluation | Unpublished |
| I* | March-20 | 855 | Lucid | COVID-19 | -Evaluation | Pennycook et al. 2020 S2[26] |
| J | April-20 | 621 | MTurk | Politics & COVID-19 | -Evaluation | Unpublished |
| K | April-20 | 444 | Lucid | Politics | -Evaluation | Unpublished |
| L | April-20 | 1192 | Lucid | COVID-19 | -Evaluation<br>-Evaluation (10x) with feedback | Epstein et al. 2021 W2[25] |
| M | May-20 | 2081 | Lucid | COVID-19 | -Evaluation<br>-Tips<br>-Norms | Epstein et al. 2021 W3[25] |
| N | May-20 | 2778 | Lucid | COVID-19 | -Tips<br>-Norms<br>-Tips+Norms | Epstein et al. 2021 W4[25] |
| O | May-20 | 2616 | Lucid | COVID-19 | -Importance<br>-Importance+Norms | Epstein et al. 2021 W5[25] |
| P | September-20 | 820 | Lucid | COVID-19 | -Evaluation-<br>-Tips | Unpublished |
| Q | September-20 | 2010 | YouGov | COVID-19 | -Evaluation<br>-Evaluation with feedback -Importance<br>+Norms<br>-PSA Video | Unpublished |
| R | September-20 | 2015 | YouGov | Politics | -Evaluation<br>-Evaluation with feedback<br>-Importance+Norms<br>-PSA Video | Guay et al. 2022[72] |
| S | November-20 | 162 | Lucid | COVID-19 | -Evaluation<br>-Tips | Unpublished |
| T | December-20 | 415 | Lucid | COVID-19 | -Tips | Unpublished |

Note that Epstein et al. (2021) consisted of four separate waves that were collected at different times, such that there were four separate instances of randomization between control and accuracy prompt conditions. We therefore treated each of those four waves as separate experiments (L–O) for the purposes of this meta-analysis. Studies that were pre-registered are indicated with *.

10% decrease relative to the meta-analytic estimate of baseline sharing intentions for false news in the control condition (intercept: $b = 0.341$, $z = 15.695$, $p < 0.001$). Conversely, there was no significant effect on sharing intentions for true news (treatment dummy from model with true as holdout for headline veracity: $b = 0.006$, $z = 1.44$, $p = 0.150$; Fig. 3). Average baseline sharing intentions was 0.341 for false headlines and 0.396 for true headlines; average sharing intentions following an accuracy prompt was 0.309 for false headlines and 0.404 for true headlines; for average sharing intentions by headline veracity and condition in each experiment, see Supplementary Information, SI, Section 1.

We find similar results when only considering the Evaluation treatment - where participants were asked to evaluate the accuracy of a single neutral headline at the outset of the study - which was the most-tested accuracy prompt ($k = 14$ experiments). The Evaluation treatment significantly increased sharing discernment (interaction between headline veracity and treatment dummies: $b = 0.034$, $z = 7.823$, $p < 0.001$), which translates into a 59.6% increase over baseline sharing discernment in the control condition (headline veracity dummy: $b = 0.057$, $z = 7.2$, $p < 0.001$). This increase in discernment was again largely driven by the Evaluation treatment significantly decreasing sharing intentions for false news (treatment dummy: $b = -0.027$, $z = -5.548$, $p < 0.001$), which translates into an 8.2% decrease relative to baseline sharing intentions for false news in the control condition (intercept: $b = 0.330$, $z = 14.1$, $p < 0.001$); the effect on sharing intentions for true news was again non-significant, $b = 0.009$, $z = 1.89$, $p = 0.059$.

To gain some insight into whether there are additive effects of exposure to multiple accuracy prompts, we compare the

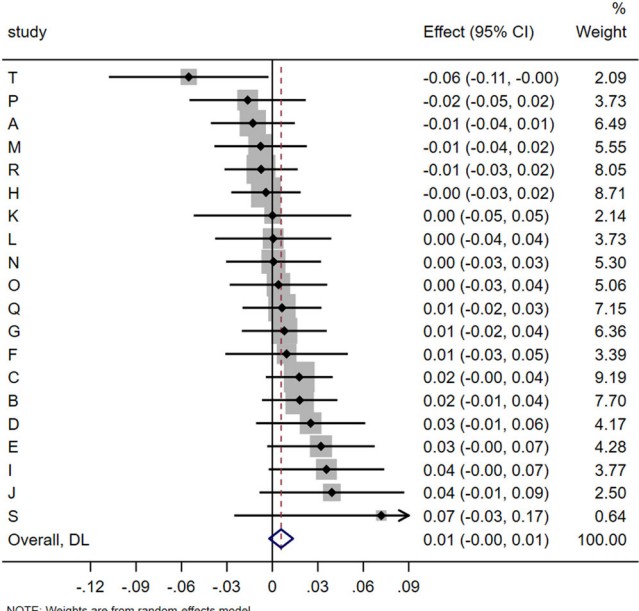

**Fig. 3 Accuracy prompts do not significantly affect sharing intentions for true news.** Meta-analytic estimate (via random effects meta-analysis) of the effect of accuracy prompts on sharing of true news across the 20 experiments analyzed in this paper. The coefficient on the condition dummy when analyzing true headlines and 95% confidence interval are shown for each study, and the meta-analytic estimate is shown with the red dotted line and blue diamond. We find no significant heterogeneity in effect size across studies, Cochran's Q test, $Q(19) = 22.42$, $p = 0.264$, $I^2 = 15.3\%$ ($k = 20$ independent studies).

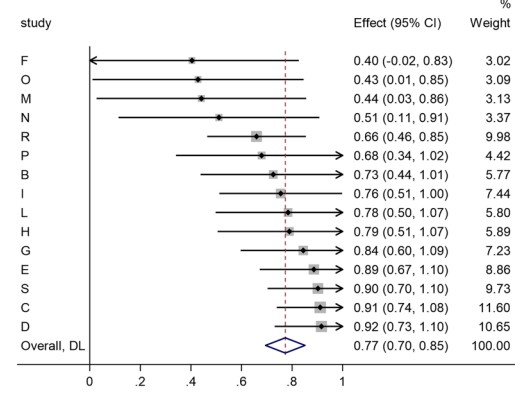

**Fig. 4 The accuracy prompt effect on sharing of a given headline is strongly correlated with that headline's perceived accuracy.** Meta-analytic estimate (via random effects meta-analysis) of the item-level correlation between the accuracy prompt effect on sharing and the headline's out-of-sample perceived accuracy rating. The correlation coefficient and 95% confidence interval are shown for each study, and the meta-analytic estimate is shown with the red dotted line and blue diamond. We find no significant heterogeneity in effect size across studies, Cochran's Q test, $Q(14) = 18.99$, $p = 0.165$ ($k = 15$ independent studies).

results for the Evaluation treatment described above to the various conditions in which the Evaluation treatment was combined with at least one additional treatment (either repeated Evaluations, indicated by "10x" in Table 2, or alternative treatments, indicated by "+" in Table 2). The combination of Evaluation and additional treatments significantly increased sharing discernment (interaction between headline veracity and treatment dummies: $b = 0.054$, $z = 2.765$, $p = 0.006$, which translates into a 100.8% increase over baseline sharing discernment in the control condition of those experiments (headline veracity dummy: $b = 0.050$, $z = 2.92$, $p = 0.003$). This increase in discernment was again largely driven by a significant decrease in sharing intentions for false news, $b = -0.048$, $z = -2.990$, $p = 0.003$, which translates into a 16.5% decrease relative to baseline sharing intentions for false news in the control condition of those experiments (intercept: $b = 0.291$, $z = 8.7$, $p < 0.001$); the effect on sharing intentions for true news was again non-significant, $b = 0.008$, $z = 0.775$, $p = 0.438$. Although not a perfectly controlled comparison, the observation that the combined treatments were roughly twice as effective as Evaluation alone suggests that there are substantial gains from stacking accuracy prompt interventions.

To test if the effect is unique to the Evaluation treatment, we examine the results when only including treatments that do not include any Evaluation elements. The non-evaluation treatments significantly increased sharing discernment (interaction between headline veracity and treatment dummies: $b = 0.039$, $z = 4.974$, $p < 0.001$), which translates into a 70.9% increase over baseline sharing discernment in the control condition of those experiments (headline veracity dummy: $b = 0.055$, $z = 7.1$, $p < 0.001$). This increase in discernment was again largely driven by

a significant decrease in sharing intentions for false news, $b = -0.039$, $z = -5.161$, $p < 0.001$, which translates into a 11.0% decrease relative to baseline sharing intentions for false news in the control condition of those experiments (intercept: $b = 0.356$, $z = 16.6$, $p < 0.001$); the effect on sharing intentions for true news was yet again non-significant, $b = 0.002$, $z = 0.338$, $p = 0.735$. Thus, the positive effect on sharing discernment is not unique to the Evaluation intervention.

**Study-level differences**. Next, we ask how the size of the treatment effect on sharing discernment varies based on study-level differences. Specifically, we consider the subject pool (convenience samples from MTurk versus more representative samples from Lucid/YouGov), headline topic (politics versus COVID-19), and baseline sharing discernment in the control condition (indicating how problematic, from a news sharing quality perspective, the particular set of headlines is).

These quantities are meaningfully correlated across studies (e.g., MTurk studies were more likely to be about politics, $r = 0.63$; and baseline sharing discernment was lower for political headlines, $r = -0.25$). Therefore, we jointly estimate the relationship between the treatment effect on discernment and all three study-level variables at the same time, using meta-regression. Doing so reveals that the treatment effect is significantly larger on MTurk compared to the more representative samples, $b = 0.033$, $t = 2.35$, $p = 0.032$; and is significantly smaller for headline sets where discernment is better at baseline, $b = -0.468$, $t = -2.42$, $p = 0.028$. (Importantly, we continue to find a significant positive effect when considering only experiments run on Lucid or YouGov, $b = 0.030$, $z = 7.102$, $p < 0.001$.) There were, however, no significant differences in the effect for political relative to COVID-19 headlines, $b = -0.017$, $t = -1.21$, $p = 0.244$.

**Item-level differences**. Next, we examine how the effect of the accuracy prompts on sharing varies across items, in a more fine-grained way than simply comparing headlines that are objectively true versus false. Based on the proposed mechanism of shifting attention to accuracy, we would expect the size of the

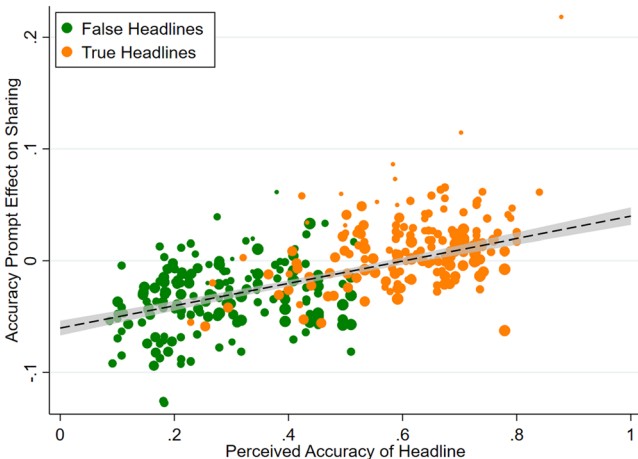

**Fig. 5 Accuracy prompts reduce sharing to the extent that headlines are perceived as inaccurate.** For each of the 357 headlines in the 15 experiments where out-of-sample accuracy ratings were available, the accuracy prompt effect (sharing when treated minus sharing in control) is plotted against the headline's perceived accuracy. False headlines are shown in green, true headlines in orange. Dot sizes are proportional to sample size. Best-fit line and 95% confidence interval are shown.

treatment effect to vary based on the perceived accuracy of the headline (since participants do not have direct access to objective accuracy). That is, to the extent that the treatment preferentially reduces sharing intentions for false headlines, the treatment effect should be more negative for headlines that seem more implausible.

In these experiments, however, participants do not rate the accuracy of each headline. Instead, for 15 experiments we have out-of-sample ratings of the average accuracy of each headline (e.g., from pre-tests, or from other experiments or conditions where participants made accuracy ratings rather than indicating sharing intentions). Thus, to assess the relationship between effect size and perceived accuracy, we conduct a headline-level analysis for each of these 15 experiments. Specifically, we correlate each headline's out-of-sample perceived accuracy with the average treatment effect for that headline (sharing in treatment minus sharing in control). Consistent with our proposed mechanism, the treatment effect is strongly correlated with perceived accuracy in all experiments (Fig. 4): the meta-analytic estimate of the correlation is $r = 0.773$, $z = 19.417$, $p < 0.001$, and the magnitude of that correlation does not vary significantly across experiments, $Q(14) = 18.99$, $p = 0.165$. To provide an additional visualization of this relationship, in Fig. 5, we pool the data across all experiments and plot the perceived accuracy and treatment effect for every headline from each experiment (weighted by sample size, $r(355) = 0.663$, $p < 0.001$).

Next, for the experiments using political headlines, we ask whether the accuracy prompts are differentially effective based on whether the headline is concordant or discordant with the participant's partisanship. Right-leaning headlines are classified as concordant for Republicans and discordant for Democrats; the opposite is true for left-leaning headlines. We find that the accuracy prompts are significantly more effective for politically concordant headlines (three-way interaction between treatment, headline veracity, and political concordance; meta-analytic estimate $b = 0.015$, $z = 3.124$, $p = 0.002$), likely because people are more likely to share politically concordant than discordant headlines at baseline (meta-analytic estimate $b = 0.102$, $z = 11.276$, $p < 0.001$). Interestingly, baseline sharing discernment

does not differ significantly for politically concordant versus discordant headlines (meta-analytic estimate $b = 0.007$, $z = 1.085$, $p = 0.278$).

We also ask whether there is evidence that the treatment effect decays over successive trials. For eight experiments (A through H), the order of presentation of the headlines was recorded; four studies had 20 trials, three studies had 24 trials, and one study had 30 trials. Examining the three-way interaction between headline veracity, treatment dummy, and trial number, the meta-analytic estimate (with a total of $N = 10,236$ subjects) is not statistically significant, $b = -0.001$, $z = -1.869$, $p = 0.062$. To the extent that there is some hint of a 3-way interaction, this is driven entirely by the first few trials. For example, when excluding the first four trials, the meta-analytic estimate of the 3-way interaction is $b = -0.000$, $z = -0.292$, $p = 0.770$ (indicating no significant order effect); and the overall treatment effect on discernment when excluding trials 1–4, $b = 0.045$, $z = 4.188$, $p < 0.001$, is very similar to the overall treatment effect on discernment when including all trials, $b = 0.047$, $z = 4.804$, $p < 0.001$. (The treatment effect on discernment in trials 1–4 was slightly larger, $b = 0.065$, $z = 5.707$, $p < 0.001$.) Similar results are observed when excluding the first five, six, seven, etc. trials; or when restricting to only the Evaluation treatment. Thus, we find evidence that the treatment effect persists at least for the length of an experimental session.

**Individual-level differences**. We now ask how the accuracy prompts' effect on sharing discernment varies within each experiment, based on individual-level characteristics. To help contextualize any such differences, we also ask how baseline sharing discernment in the control varies based on each individual-level characteristic. Furthermore, because the distribution of the individual-level variables is extremely different for samples from MTurk (which makes no attempt at being nationally representative) versus the more representative samples from Lucid or YouGov (see Supplementary Information, SI, Section 1 for distributions by pool), we conduct all of our individual-level moderation analyses separately for Lucid/You-Gov versus MTurk – and in our interpretation, we privilege the results from Lucid/YouGov, due to their stronger demographic representativeness. The results for all measures are summarized in Table 3.

We begin with demographics, which are of broad interest for misinformation research because differences in effectiveness across demographic categories has important implications for the targeting of interventions. With respect to gender, we find no significant moderation in either the more representative samples or MTurk. Interestingly, women are significantly less discerning in the control condition on MTurk, but not in the more representative samples. With respect to participants who identified as white versus other ethnicity or racial categories, we also find no significant moderation in either set of subject pools, and no significant differences in baseline discernment. With respect to age, we find that the accuracy prompts have a significantly larger effect for older participants in the more representative samples but not MTurk, while older participants are more discerning in their baseline sharing on both platforms (we find no evidence of significant non-linear moderation by age when including quadratic age terms). With respect to education, we find in both sets of samples that the accuracy prompts had a larger effect for college educated participants, and that college educated participants were more discerning in their baseline sharing. Importantly, however, the accuracy prompts still improve sharing discernment even for non-college educated

**Table 3 Coefficients, z values, and two-tailed p values for the individual-level difference moderation analyses, derived from the linear regression models described in the text.**

| | Moderation of accuracy prompt effect on discernment | | | | | | Relationship with baseline discernment in control | | | | | |
| --- | --- | --- | --- | --- | --- | --- | --- | --- | --- | --- | --- | --- |
| | Lucid/YouGov | | | MTurk | | | Lucid/YouGov | | | MTurk | | |
| | b | z | p | b | z | p | b | z | p | b | z | p |
| Female | −0.006 | −0.814 | 0.416 | −0.005 | −0.380 | 0.704 | −0.001 | −0.160 | 0.873 | **−0.019** | **−2.175** | **0.030** |
| White | 0.014 | 1.746 | 0.081 | −0.011 | −0.773 | 0.440 | 0.006 | 0.966 | 0.334 | 0.042 | 1.899 | 0.058 |
| Age | **0.001** | **3.182** | **0.001** | 0.000 | 0.840 | 0.401 | **0.001** | **5.661** | **<0.001** | **0.001** | **2.051** | **0.043** |
| College degree | 0.001 | 0.139 | 0.890 | −0.011 | −0.919 | 0.358 | **0.037** | **5.116** | **<0.001** | **0.030** | **2.371** | **0.018** |
| Conservatism | 0.009 | 0.623 | 0.533 | −0.031 | −1.693 | 0.090 | **−0.120** | **−4.531** | **<0.001** | **−0.069** | **−2.220** | **0.026** |
| Republican | −0.007 | −1.410 | 0.158 | **−0.022** | **−3.057** | **0.002** | **−0.070** | **−4.209** | **<0.001** | −0.022 | −1.270 | 0.204 |
| Republican (no independents) | −0.005 | −0.867 | 0.386 | **−0.022** | **−2.526** | **0.012** | **−0.086** | **−4.018** | **<0.001** | −0.036 | −1.781 | 0.075 |
| Trump 2016 voter | −0.001 | −0.087 | 0.931 | **−0.027** | **−2.744** | **0.006** | **−0.053** | **−3.031** | **0.002** | −0.014 | −0.803 | 0.422 |
| Value accuracy | 0.004 | 0.315 | 0.753 | **0.028** | **2.755** | **0.006** | **0.097** | **6.404** | **<0.001** | **0.087** | **7.449** | **<0.001** |
| Cognitive reflection (CRT) | **0.038** | **1.986** | **0.047** | **0.030** | **2.371** | **0.018** | **0.062** | **5.351** | **<0.001** | **0.061** | **4.117** | **<0.001** |
| Attentiveness | **0.068** | **4.090** | **<0.001** | | | | **0.067** | **7.628** | **<0.001** | | | |

The left half of the table shows the coefficients for the 3-way interaction between headline veracity, condition, and the individual difference - which captures the extent to which the individual difference moderates the treatment effect on sharing discernment. The right half of the table shows the coefficient on the 2-way interaction between headline veracity and the individual difference - which captures how the individual difference relates to baseline sharing discernment in the control condition. Results are shown separately for the more representative samples from Lucid or YouGov, versus the convenience samples from MTurk. Coefficients with $p < 0.05$ are bolded. No adjustments were made for multiple comparisons.

participants (Lucid/YouGov, $b = 0.033$, $z = 4.940$, $p < 0.001$; MTurk, $b = 0.059$, $z = 3.404$, $p = 0.001$).

Next, we turn to political orientation (see Fig. 6). We find no significant moderation of the accuracy prompt effect by conservative (relative to liberal) ideology, either in the more representative samples or on MTurk. Importantly, we do find a significant negative relationship between conservatism and baseline sharing discernment (i.e., overall quality of news that is shared) in the control, and this is evident in both sets of subject pools (although the relationship is twice as large in the more representative samples). This worse baseline discernment aligns with real-world sharing data where conservatives were found to share more fake news on Twitter[10] and Facebook[12]. This suggests that the lack of moderation by conservatism is not due to potential limitations of our conservatism measure.

To shed further light on the potential moderating role of ideology, we also re-analyze data from a Twitter field experiment[24] in which an overwhelmingly conservative set of users who had previously shared links to Breitbart or Infowars – far-right sites that were rated as untrustworthy by fact-checkers[41] – were prompted to consider accuracy. Note that users' ideology had been estimated based on the accounts they follow[42], but moderation of the treatment effect by ideology was not assessed previously. Critically, the pattern of results for this Twitter data matches what we report above for the survey experiments: Users who are more conservative shared lower quality information at baseline, but we do not find evidence that ideology moderated the effect of the accuracy prompts. See SI Section 3 for analysis details.

Returning to our meta-analysis of the survey data, as a robustness check we also consider partisanship, rather than ideology, using three binary partisanship measures (preferred party, party membership excluding Independents, and having voted for Donald Trump in the 2016 U.S. Presidential Election). In the more representative samples, across all three measures we find that people who identify as Republican were significantly less discerning in their baseline sharing (in line with what has been observed previously), but partisanship does not significantly moderate the accuracy prompt effect on discernment. On MTurk, we see the opposite pattern: Republicans were not significantly less discerning in their baseline sharing, yet partisanship does significantly moderate the accuracy prompt on discernment, with the prompts working less well for Republicans (although the prompts still significantly increase sharing discernment among Republicans on MTurk, however defined: participants who prefer the Republican Party, $b = 0.037$, $z = 2.988$, $p = 0.003$; participants who identify with the Republican Party, excluding Independents, $b = 0.035$, $z = 2.790$, $p = 0.005$; participants who voted for Trump in 2016, $b = 0.032$, $z = 2.608$, $p = 0.009$; similar patterns are observed with only considering the Evaluation treatments, see SI Section 3. Although we have comparatively low power for statistically analyzing heterogeneity across studies (16 to 20 studies, depending on the individual difference), for completeness in SI Section 5, we report the results of meta-regressions predicting moderation using platform type, news type, and baseline discernment. The difference across platforms in the moderating effect of partisanship on the treatment effect was significant for people who voted for Trump versus those who did not ($p = 0.033$), marginally significant for participants' preference for the Democratic versus Republican party ($p = 0.078$), and not significant for Democratic versus Republican party membership ($p = 0.198$); although these results should be interpreted cautiously in light of low statistical power.

The explicit (self-reported) importance participants placed on accuracy when deciding what to share did not moderate the treatment effect on sharing discernment in the more representative samples, but positively moderated the treatment effect on

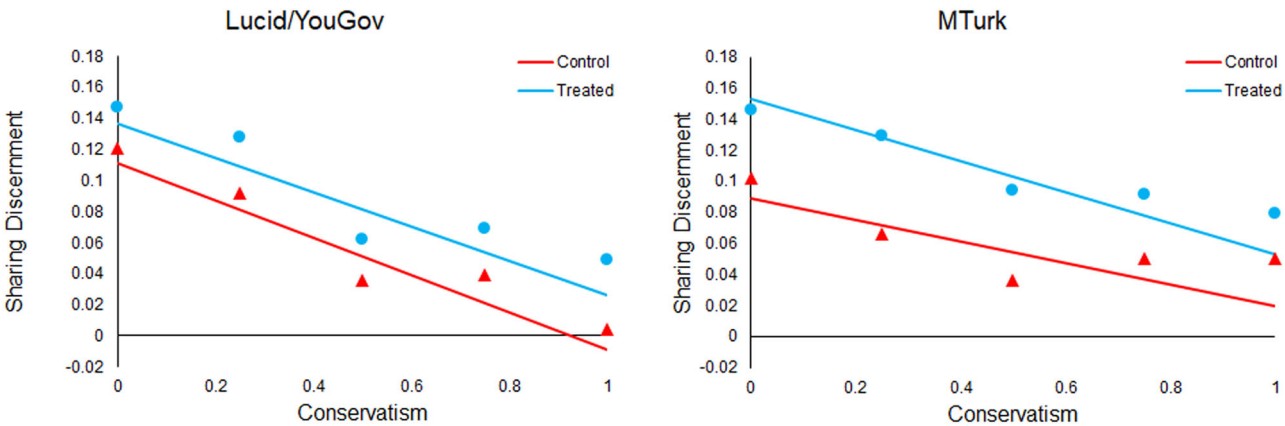

**Fig. 6 Accuracy prompts increase sharing discernment across the ideological spectrum.** Shown is sharing discernment in the control (red triangles) versus treatment (blue circles) as a function of liberal versus conservative ideology. The model fits for discernment in control and treatment, based on meta-analytic estimates of model coefficients, are shown with solid lines. The meta-analytic estimate of discernment in control and treatment at each level of conservatism (rounded to the nearest 0.25) are shown with dots. More representative samples from Lucid and YouGov are shown in the left panel; convenience samples from Amazon Mechanical Turk are shown in the right panel.

MTurk; and was positively associated with baseline sharing discernment in both types of subject pool.

When it comes to performance on the Cognitive Reflection Test, the more representative samples and MTurk show similar results. In both cases, participants who score higher on the CRT show a larger effect of the accuracy prompts, and are also more discerning in their baseline sharing. Nonetheless, the accuracy prompts still increase sharing discernment even for participants who answer all CRT questions incorrectly (more representative samples, $b = 0.029$, $z = 2.784$, $p = 0.005$; MTurk, $b = 0.034$, $z = 2.119$, $p = 0.034$; combined, $b = 0.030$, $z = 3.581$, $p < 0.001$).

Lastly, we examine the association with attentiveness in the 8 studies (all run on Lucid) that included attention checks through the study. An important caveat for these analyses is that attention was measured post-treatment, which has the potential to undermine inferences drawn from these data[43]. Keeping that in mind, we unsurprisingly find that the accuracy prompts had a much larger effect on participants who were more attentive, and that more attentive participants were more discerning in their baseline sharing. In fact, we find no significant effect of the accuracy prompts on sharing discernment for the 32.8% of participants who failed a majority of the attention checks, $b = 0.007$, $z = 0.726$, $p = 0.468$, while the effect is significant for the 67.2% of participants who passed at least half the attention checks, $b = 0.039$, $z = 4.440$, $p < 0.001$.

## Discussion

Here, we provide evidence, across 20 experiments with U.S. samples, that a variety of accuracy prompts robustly increased social media users' sharing discernment by decreasing their intentions to share a wide range of false news headlines about politics and COVID-19. Furthermore, the effect was strongest for headline sets that were the most challenging (i.e., where baseline sharing discernment was lowest), generalized across demographic groups and samples, was not moderated by self-reported political ideology (or partisanship in our more nationally representative samples), and persisted over the course of the experimental session.

The replicability and generalizability demonstrated here – together with a previously reported Twitter field experiment demonstrating efficacy on platform[24] – suggest that accuracy prompts are a promising approach for technology companies, governments, and civil society organizations to investigate in their efforts to reduce the spread of misinformation. Of course, no

single approach will solve the misinformation problem. Accuracy prompts should be therefore considered in combination with a wide range of other approaches[14,44]. Moreover, the effects we document here, while being replicable and generalizable, are modest in size (although it is unclear how the magnitude of effects observed in the survey experiments we conducted here relate to the actual effect sizes that would be observed on platform, especially given the possibility of network effects that amplify individual-level effects; see Ref. [24] SI Section 6 for illustrative network simulations). If technology companies explore accuracy prompt interventions, they should conduct experiments to optimize the treatment format and delivery, with the goal of maximizing treatment effect sizes and durability.

In addition to these clear practical implications, our meta-analytic results also have numerous theoretical implications. First, we provide evidence of a robust positive association between performance on the Cognitive Reflection Test (CRT) and sharing discernment. Prior work indicates that cognitive reflection is positively associated with truth discernment when making accuracy judgments[18,45]. Showing that this relationship also occurs for sharing decisions is an important extension, given the dissociation between accuracy judgments and sharing intentions observed in past work[24–26]. Future work should investigate the extent to which the correlation with sharing discernment is driven by individuals who score higher on the CRT having more accurate underlying beliefs, versus being more averse to sharing falsehoods (or both). Interestingly, the observation that the accuracy prompts are more effective for higher CRT individuals speaks against the correlation with sharing discernment being solely explained by higher CRT individuals being more attentive to accuracy at baseline (although more direct evidence is necessary). Future work should also attempt to resolve apparent inconsistencies between studies correlating CRT with actual sharing on social media[27,46]; for example, by accounting for what content users were exposed to, rather than simply examining what they share.

Second, the results provide additional support for the claim that inattention to accuracy contributes to misinformation sharing and that the accuracy prompts are effective because they draw attention back to accuracy[24,26]. Consider the observation that the treatment effect is smaller for headline sets where baseline discernment is better (and, therefore, that the treatment effect is larger for headline sets where baseline discernment is worse). One possibility that is consistent with the inattention account is that

baseline sharing discernment is worse in cases where the content is particularly distracting (e.g., it is particularly emotional[47], or it contains moral content[48,49]). Therefore - as we find - it should be these headline sets where accuracy prompts are most effective. In contrast, if the problem driving poor discernment was primarily a more fundamental confusion about which headlines were true versus false, then we would expect the opposite pattern: shifting attention to accuracy should be less effective for headlines where people are more confused about the headlines' accuracy.

Furthermore, the similar effect size for political versus COVID-19 headlines suggests that the accuracy prompt effects are operating through a process that is general across content types (such as inattention), rather than through a process that is specific to one type of news or the other. This logic also has implications for the effects of accuracy prompts on true headlines. In the experiments analyzed here, the true headlines were typically fairly ambiguous - see, for example, pre-test data[24] and accuracy-only conditions[26] in recent accuracy prompt studies, where only roughly 60% of true headlines were rated as accurate. Thus, although the accuracy prompts had no significant effect on these headlines, we would expect them to significantly increase sharing of more unambiguously true headlines (see item-analyses in Refs. [24–26]).

These findings highlight the importance of understanding the nature of the content used in studies on misinformation, as well as the characteristics of the individuals who are being targeted with the intervention. For an accuracy prompt to have an effect on sharing, the individual in question has to be able to recognize, to some extent, that the content in question is seemingly false (for them to decide not to share it) or true (for them to decide to share it). Thus, if a group of participants is targeted who have a very difficult time distinguishing between true and false content, or if misinformation is identified that the majority of a particular group of people believe to be true, then accuracy prompts will not improve sharing discernment. We suggest that studies investigating the efficacy of accuracy prompts take this consideration into account and include a separate condition that explicitly measures the believability of the content that they are employing for individuals who are similar to those that are being tested.

More broadly, these results emphasize tradeoffs between different subject pools. On the one hand, we found that the treatment effect sizes were larger on MTurk than when using more representative samples from Lucid or YouGov. This is likely due in large part to participants on MTurk being quite a bit more attentive than participants on Lucid[50], and unsurprisingly, we found that the treatment was ineffective for Lucid participants who failed a majority of attention checks. On the other hand, however, MTurk offers convenience samples and makes no attempt to be nationally representative (and is, in fact, not representative, particularly when it comes to self-reported political identity and ideology[51–53]). As a consequence, individual difference results - particularly related to politics - from MTurk can be misleading. For example, while data from the more representative samples replicated previously observed findings where people who identify as Republican share more fake news than those who identify as Democrats, the MTurk data showed the opposite pattern, failing to find any baseline relationship between partisanship and sharing discernment. Contrary to past work assessing MTurk that focused on personality and values[54], our data indicates that researchers should avoid using MTurk samples to make strong claims about differences between Democrats and Republicans.

Our findings may also have relevance for a major recent debate about the replicability of so-called "social" or "behavioral" priming effects[55]. This form of priming is said to occur when "a stimulus influences subsequent behavior without conscious guidance or intention"[55]. Several prominent examples of behavioral priming have proven difficult to replicate[56,57] and scholars have argued that replicable priming effects are restricted to "perceptual primes" (e.g., as in semantic priming), which involve (for example) perceiving an ambiguous stimuli in a way that is more consistent with a prime[58]. Within the context of this debate, some may view the accuracy prompt effect as an example of a highly replicable between-subject behavioral (non-perceptual) priming effect: This meta-analysis clearly shows that increasing the salience of the concept of accuracy has downstream consequences on sharing intentions. However, whether this effect is operating outside of conscious awareness or intentionality has not yet been tested. This is an interesting area for future research that will help illuminate whether accuracy prompts are indeed a replicable example of behavioral priming.

Finally, we consider limitations and additional potential directions for future work on accuracy prompts. Although the benefit of internal meta-analyses is that one can be confident that the entire "file-drawer" is included and that p-hacking has been avoided, the obvious downside is that the approach does not take other relevant (external) studies into account. While the one replication of an accuracy prompt intervention[29] we are aware of that meets our inclusion criteria (conducted in the U.S. between 2017 and 2020) did find a significant positive effect on sharing discernment – and including this study in our main analysis does not meaningfully alter the results (see SI Section 4) – it would be valuable for future work to examine studies conducted by a wider range of research groups. Furthermore, given the global nature of the misinformation challenge, it is essential to test how the findings from U.S. participants generalize across cultures[59,60]. Indeed, a recent cross-cultural experiment across 16 countries found broad evidence of replicability for accuracy prompts, and found evidence that the effect was largest in countries where participants had a greater disconnect between accuracy and sharing[61].

More data from field experiments is also critical. This could include experimenting with alternative methods for delivering accuracy prompts, for example via advertisements or public posts, as well as experimenting on platforms other than Twitter, such as YouTube or Facebook. Future work should also provide a more detailed mechanistic account of how the accuracy prompts increasing sharing discernment. This could include investigating what, precisely, causes people to be inattentive to accuracy (e.g., are there characteristics that cause more distraction when reading particular headlines, do things differ from platform to platform, etc.). Computational modeling, for example using limited-attention utility models[24] or drift-diffusion models[62], could also play an important role in such investigations. Lastly, it would be fruitful to explore how accuracy nudges interact with other approaches to fighting misinformation, such as labeling false content[63–65] or increasing media and digital literacy[19,60,66].

In sum, the meta-analysis presented here suggests that accuracy prompts may be a promising addition to social media platforms' anti-misinformation toolkit. Accuracy prompts have the potential to help improve the quality of news shared online in a scalable way, without undermining user autonomy or making platforms the arbiters of truth.

## Methods

**Study selection**. For our internal meta-analysis, we searched our internal lab records to identify raw data for studies satisfying the following inclusion criteria: (a) study conducted between 2017 and 2020; (b) subject pool from the U.S.; (c) participants completed the study via an online survey; (d) participants were asked to indicate their likelihood of sharing a set of true and false headlines, the veracity of which were rated by professional fact-checkers; and (e) participants were randomized to a control condition or one or more treatments in which an accuracy prompt was administered prior to the sharing task. We also had one exclusion criterion: studies in which participants also rated the accuracy of each headline, as

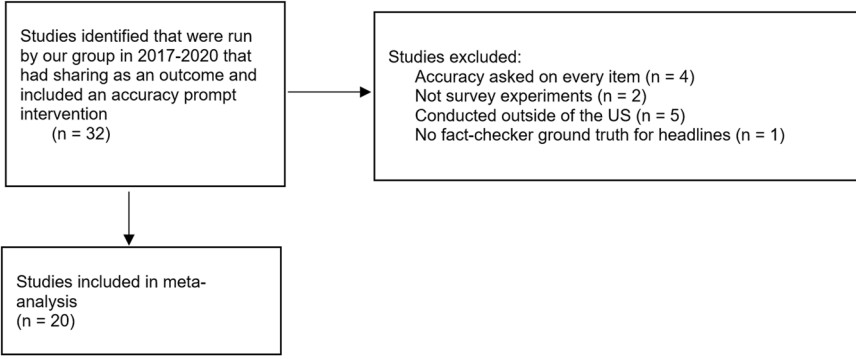

**Fig. 7 Flow diagram for study selection.** Demonstration of study selection for our internal meta-analysis.

well as indicating sharing intentions, were ineligible because asking jointly about accuracy dramatically affects sharing intentions[24]. Although these inclusion and exclusion criteria were determined prior to running the meta-analysis, the meta-analysis was not pre-registered and no protocol was prepared.

As shown in Fig. 7, of 32 possible studies, four experiments were excluded because rather than receiving an accuracy prompt at the outset, participants in the treatment were asked to rate the accuracy of every single item immediately prior to making their sharing decision (a "full attention treatment"); two experiments were excluded because they were conducted using an in-development platform that more closely simulates a social media feed rather than in a survey (these studies were pilots during which the platform was being debugged and thus the data are not usable); five experiments were excluded because they used non-US participants (we are exploring cross-cultural variation in a separate project[61]); and one experiment was excluded because third-party fact-checker ratings were not available for the headlines, which prevents us from assessing treatment effects on the veracity of information sharing. Similarly, one study that was included had, in addition to true and false headlines, headlines that were hyperpartisan, and these headlines were excluded when analyzing that study for the meta-analysis. Both authors agreed on the classification of each study as eligible versus ineligible. In total, our inclusion and exclusion criteria therefore yielded $k = 20$ eligible experiments with a total of $N = 26,863$ participants. Table 2 summarizes each study, and study-level data and code required to reproduce all reported analyses are available at OSF.

**Experimental designs**. In each study, only participants who indicated that they use social media were allowed to participate (there were no inclusion or exclusion criteria based on the types of content people reported sharing online – all participants indicated informed consent). Participants were presented with a set of actual true and false news headlines taken from social media one at a time and in a random order (these were presented in the format of a Facebook post; see Ref. [67] for a detailed explanation of the methodology behind headline selection). All of the false headlines were found using popular fact-checking sites such as snopes.com and factcheck.org and all of the true headlines came from reputable mainstream news sources. In most cases, headlines were selected for inclusion based on prior pre-testing; for all experiments using political headlines, the headlines were balanced on partisan lean based on the pre-test results (i.e., there were both Pro-Democratic and Pro-Republican headlines, and they were equally partisan based on pre-test ratings). Furthermore, there was an attempt in each case to select headlines that were "up to date" or relevant for when the study was run.

As detailed in Table 2, key dimensions of variation across included studies were the subject pool from which the participants were recruited (convenience samples from Amazon Mechanical Turk; samples from Lucid that were quota-matched to the national distribution on age, gender, ethnicity, and region; or samples from YouGov that use sample matching to select representative samples from non-randomly selected pools of respondents[68]), the topic of the headlines about which the participants were make sharing decisions (politics versus COVID-19), the specific set of headlines shown (and thus the baseline level of sharing discernment between true versus false headlines), and the particular set of accuracy prompts employed (see Table 1 for a description of each accuracy prompt). We examine how the effect of the accuracy prompts varies across these dimensions.

**Individual difference measures**. In addition to study-level variables, we also examine how various individual-level variables moderate the effect of the accuracy prompts. With respect to basic demographics, we examine age, gender, and education (0 = less than a college degree, 1 = college degree or higher), which were collected in all 20 experiments, and race/ethnicity (coded as 1 = identified as white, 0 = did not identify as white), which was collected in 17 experiments. We also examine political orientation. In doing so, we focus on liberal versus conservative ideology, because it was collected in all 20 experiments using Likert scales (rather than binary or categorical variables), thus providing a more nuanced measure than partisanship (i.e., Democrat versus Republican). Specifically, in 16 experiments,

participants were asked separately about how socially and economically liberal versus conservative they were, using 7-point Likert scales; we average the two items to generate an overall liberal versus conservative measure. In 2 experiments (Q and R), participants were asked a single question about how liberal versus conservative they were, using a 5-point Likert scale. In 2 experiment (S and T), participants were asked the extent to which they thought incomes should be made more equal versus there should be greater incentives for individual effort, and the extent to which they thought government should take more responsibility to ensure that everyone is provided for versus people should take more responsibility to provide for themselves, using 10-point Likert scales, and we average the two items to generate a liberal versus conservative measure. In all experiments, the final measure was then rescaled to the interval [0,1].

As robustness checks, we also examine a binary measure of preference for the Democratic versus Republican party (forced choice, no neutral option), which was collected in 18 experiments. In 5 experiments, this question was asked as a binary forced choice. In 13 experiments, it was asked as a 6 point Likert scale (no neutral option) and then binarized for analysis. In 2 experiments (Q and R), it was asked using a 7-point Likert scale; participants who chose the neutral middle option were asked which party they preferred and categorized accordingly; and the 0.3% of respondents who insisted that they did not prefer either party were excluded. Furthermore, we consider a binary measure of whether or not participants reported voting for Donald Trump in the 2016 U.S. Presidential Election, which was collected in 17 experiments.

Further, we examined participants' propensity to stop and think rather than going with their gut response, as measured by the Cognitive Reflection Test (CRT[69,70]). The CRT involves asking participants a series of questions with intuitively compelling but incorrect answers and was collected in 17 experiments. The number of CRT questions used varied between 3 and 7 across experiments. For all analyses, we used the fraction of questions answered correctly. We also examined the importance that participants self-reported placing on only sharing accurate news on social media (as per[24]), which was measured in 17 experiments. Finally, attention check questions were included throughout the study (as per[71]) in 8 experiments. (Many studies also prevented respondents who failed trivial attention checks at the study outset from participating, but since no data was collected from these participants, we cannot assess the impact of extreme inattention.)

**Analysis approach**. For analysis purposes, sharing decisions (the dependent variable) are rescaled such that the minimum possible value is 0 (indicating a very low likelihood of sharing) and the maximum possible value is 1 (indicating a very high likelihood of sharing). Within each study, we conduct a rating-level (i.e., one observation per subject-item pair) linear regression with robust standard errors clustered on participant and headline, taking sharing intention as the dependent variable. Our main analysis includes a dummy for headline veracity (0 = false, 1 = true), a dummy for condition (0 = control, 1 = accuracy prompt), and the interaction term. With this specification, the coefficient on the interaction term indicates the accuracy prompt's effect on sharing discernment (the difference in sharing likelihood for true relative to false headlines), the coefficient on the condition dummy indicates the accuracy prompt's effect on sharing intentions for false headlines, and the coefficient on the headline veracity dummy indicates baseline sharing discernment in the control condition. Our participant-level heterogeneity analyses use models that add the individual difference being interrogated along with all interaction terms, and focus on the 3-way interaction (the extent to which the individual difference moderates the accuracy prompt's effect on sharing discernment); and our analysis of order effects adds trial number along with all interaction terms, and again focuses on the 3-way interaction.

For any given coefficient of interest, we calculate an estimate for each study. Our interest, however, is not the effect in any given study. Instead, we are interested in the best estimate of the effect using the data from all studies. Therefore, for each coefficient of interest, we combine the estimates from each study using random effects meta-analysis to generate this overall estimate – what we refer to as the

"meta-analytic estimate" of the value of that coefficient. We use random effects meta-analysis, rather than fixed-effects meta-analysis, because there is reason to expect that the true effect size varies across studies (because, for example, different studies used different versions of the treatment, different headlines, and different subject pools).

**Reporting summary**. Further information on research design is available in the Nature Research Reporting Summary linked to this article.

## Data availability
The study-level data for this meta-analysis (along with the code) is available on OSF: https://osf.io/4mv9z/.

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

## Acknowledgements

We gratefully acknowledge funding from the Ethics and Governance of Artificial Intelligence Initiative of the Miami Foundation (G.P., D.R.), the William and Flora Hewlett Foundation (D.R.), the Reset Initiative of Luminate (part of the Omidyar Network) (G.P.), the John Templeton Foundation (G.P., D.R.), the TDF Foundation (D.R.), the Canadian Institutes of Health Research (G.P.), the Social Sciences and Humanities Research Council of Canada (G.P.), the National Science Foundation (D.R.), and Google (G.P., D.R.). Additional materials for this manuscript can be found on OSF: https://osf.io/4mv9z/.

## Author contributions

G.P. and D.R. developed the study concept and contributed to the study design. The manuscript was written by both G.P. and D.R. Data analysis was completed by D.R.

## Competing interests

G.P. and D.R. have received research funding from Google, and D.R. has received research funding from Meta.
