## [Peer Review File · Nature Communications]

Title: Accuracy prompts are a replicable and generalizable approach for reducing the spread of misinformationREVIEWER COMMENTS

Reviewer #1 (Remarks to the Author):

In this ms., the Authors conduct a meta-analysis of 20 of their own studies bearing on whether accuracy prompts (i.e. reminding people of the importance of sharing accurate information, in different ways) increase sharing discernment (i.e. the sharing of true news by compared to fake news). They find that the effects of accuracy prompts are very robust and generalizable, and derive theoretical implications from their results.

This is a solid ms., and we (this is a joint review) think that the ms. should be accepted down the line. We do have a number of comments.

On the introduction, we believe the Authors do not do a very good job summarizing the existing literature. First, the opening paragraph is misleading. The Authors state, rightly, that fake news have received much attention in the press and the scientific literature. Many readers will likely infer from that that this is because fake news is a significant problem. Instead of implying this conclusion, and talking about the 'meta' level (i.e. about the research, rather than the phenomenon), it would be better if the Authors could succinctly lay out the main conclusions of that research (i.e. regarding the reach of fake news, the likelihood of it affecting elections, etc.).

Later on, the Authors are somewhat unfair in their presentation of different theories explaining why people share fake news. When discussing others' theories, they present them as factors that "may also contribute to misinformation sharing," while "evidence suggests that mere inattention to accuracy [the Authors' theory] plays an important role." This is not a fair representation of the existing evidence, some of which suggests that political partisanship is the main driver of sharing fake news, while other factors play a more minor role (see for instance the Osmundsen paper cited by the Authors).

In that same paragraph, the Authors write that "purposeful sharing of falsehoods is relatively rare." It's not entirely clear what the Authors mean by "purposeful sharing of falsehoods." For instance, if I share a piece of news even though I'm unsure about its accuracy, because it has other qualities (e.g. being provocative), am I purposefully sharing a falsehood? If I share something from the Onion? If I share something that I caveat? Maybe it would be better to remove that statement, since it's hard to imagine there'll be enough space to clarify and defend it here.

Turning to the results, they appear on the whole to be very solid, and some of the figures (e.g. Fig 5) are very informative. A few notes:

A recent paper pointed out some potential limitations of internal meta-analyses

http://urisohn.com/sohn_files/wp/wordpress/wp-content/uploads/ima-internal-meta-analysis-99-published-edited.pdf

The present meta-analysis should be exempt from most of these pitfalls (e.g. all the study were pre-registered) but not from others (e.g. the decision to include the studies in the meta-analysis was likely decided after having run these studies). We don't believe that the violation of some of these criteria make the present meta-analysis useless, but they should be discussed.

"In each study, only participants who indicated that they use social media were allowed to participate." Were participants saying that they don't share news on social media also excluded?

Were most answers recorded on scales, and then recoded as dichotomous outcomes? Could that affect the results in any way?

Although the existing figures are helpful, it would be nice to have more descriptive data. As it stands, the reader has no idea of what proportion of people share false vs. true news (with or without the intervention). We believe that such descriptive data is really important to get a fuller understanding of a phenomenon.

Regarding the theoretical implications:

The Authors note that "Consider the observation that the treatment effect is smaller for headline sets where baseline discernment is better (and, therefore, that the treatment effect is larger for headline sets where baseline discernment is worse). One possibility that is consistent with the inattention account is that baseline sharing discernment is worse in cases where the content is particularly distracting" First, a possibility is that when baseline discernment is worse there is simply more room for improvement, and thus that this result might be largely artefactual. Second, the finding, if not artefactual, is compatible with other theories, for instance if people were purposefully sharing misinformation.

"Researchers should avoid using MTurk data to 424 make strong claims about differences between Democrats and Republicans." This is true, but the Authors might not be the first to make that claim—if that's the case, the Authors' case would be bolstered by citing previous research.

"This could include investigating what, precisely, causes people to be inattentive to accuracy (e.g., are there characteristics that cause more distraction when reading particular headlines, do things differ from platform to platform, etc.), and what people are attending to instead of accuracy." Phrasing the question as "what people are attending to instead of accuracy" is misleading. It's not as if people could only attend to one feature of a piece of information at a time. A better question is: what are the other factors people pay attention to, besides accuracy, when consuming and sharing information. This broad question has been investigated by many disciplines (e.g. use and gratification theory in media studies, cultural attraction theory in cultural evolution, relevance theory in pragmatics, etc.). So the Authors are suggesting here to do something that a great many scholars have been doing for many years.

typos

“are significant more effective”

“is strong correlated”

“(e.g., it is particularly emotional²⁹, or contains moral content^{30,31}.”

Missing parenthesis

“in worse” should be is worse

Reviewer #2 (Remarks to the Author):

Review of Nature Communications manuscript NCOMMS-21-40911

Shifting attention to accuracy is a replicable and generalizable approach for reducing the spread of fake news

The manuscript aims to provide a meta-analytic review of evidence behind a range of accuracy prompts. Such prompts can be used to slow the spread of online misinformation without imposing any content restrictions. As such, this is a useful and important contribution to the field, which however at this stage of writing & research suffers from a few issues, which need to be addressed in the revision.

Major points

1. First of all, this is not really a proper meta-analysis of accuracy prompts but of a subset of studies done in one lab. Thus, the study is not following the standard meta-analytic procedure where there is a protocol for inclusion of the studies in the analysis. This might not be a problem per se, since the goal of this particular paper is narrower.

However, I do want to raise a general concern of a potential bias due to the fact that the authors of the meta-analysis and all the studies are the same.

Relatedly, the authors do not engage with research outside of their own lab. For example, pre-registered replication of one of their study by Roozenbeek et al (2021) is not even mentioned in the current manuscript. This is surprising, as I think critical engagement with relevant research on accuracy prompts done outside of the authors' lab is necessary at this stage of evidence evaluation.

Moreover, the authors themselves write “There are two main threats to the validity of meta-analytic results: the systematic omission of studies and the flexible selection of analysis approaches within each study inflating the rate of false positives”. However, it seems that systematically focusing only on their own studies and not even conducting a systematic search for study inclusion, they contradict

themselves.

2. Second point concerns conceptual and theoretical underpinning of this research.

Here, a better conceptual explanation as well as concrete description of Accuracy prompts included in the meta-analysis is needed. Table 1 includes different types of interventions subsumed under the “accuracy prompt” umbrella, such as reminders, social norms, nudges and even media literacy tips. They all indeed might have to do with the concept of accuracy but engage participants in fundamentally different ways. I would suggest starting with a definition of what the accuracy prompt means, then outline how it can be enacted in different ways and through different cognitive mechanisms and then how it can be implemented experimentally and in the social media environment. I had to go to the OSF to dig up experimental stimuli to actually see what these different interventions are about – unfortunately, I only found the video and none of the other stimuli. Thus it is really impossible to see what the “Tips” prompt is about (described in the table merely as “Participants are shown a set of minimal digital literacy tips.”) So far, I am not convinced that accuracy prompts and digital literacy tips can be subsumed under the same umbrella – but more information is needed and I think this information should be provided in the paper itself and/or made accessible in the supplement.

I would also like to see at least some explanation of the dependent variable (sharing discernment): why is it of central interest for interventions research (e.g., as opposed to truth discernment)? How is it related to the goal of reducing spread of misinformation and so on.

3. According to the editor’s request, my role was not to evaluate validity and reproducibility of statistical analyses (I am neither an expert in meta-analyses nor do I work with Stata), and I hope another reviewer(s) will do so. Their opinion should also have more weight on this matter. That being said, my impression is that the methodology lacks transparency and could be presented in a more clear and reproducible way. For instance, I would suggest to expand the “Analysis approach” section adding more details on the meta-analytic approach, where the authors could explain their choice of the random effects meta-analysis, discuss heterogeneity of effects, and explain in more detail the summary effects and their main quantity of interest (what they call the meta-analytic estimate). Same goes for all constitutive parts of meta-analytic review process itself (such as inclusion criteria). Here might be the place to discuss your decision not to engage with studies done outside of your lab).

In the Results, when reporting main effects (e.g., on p. 5 “We find that accuracy prompts significantly increase sharing discernment (interaction between headline veracity and treatment dummies; Figure 1), $b = 0.038$, $z = 7.102$, $p < .001$, which translates into a 71.7% increase over the meta-analytic estimate of baseline sharing discernment in the control condition, $b = 0.053$, $z = 6.636$, $p < .001$.”), please specify how you have arrived to this number. In general, I think reporting relative numbers (e.g., increase in 72%) tends to inflate the actual effects. Why not report percentage points instead?

It would also be helpful to report these numbers for control and treatment conditions in the plot (as it is sometimes done in forest plots reporting meta-analytic results), or at least include them in the table

format in the SI (along with other relevant statistics, such as heterogeneity of effects in included studies). As a general rule, if a number is reported in the paper it should be supported by a plot or a results table either in the main text or in the supplement.

Minor points

In Methods, p. 20, “convenience samples from Amazon Mechanical Turk; samples from Lucid that were quota-matched to the national distribution on age, gender, ethnicity, and region; or representative samples from YouGov” – in what way YouGov samples were nationally representative? Were they also quota-matched?

There are several other minor points, but I do not think it is useful to engage with them until the major points are addressed.

Reviewer #3 (Remarks to the Author):

I was asked in particular to comment on the meta-analytic approach that the authors took in this paper. I love what the authors have done here – this is exactly the kind of approach that psychologists should be taking with their work in general: presenting all of the evidence they have to test a particular effect. And as the authors note, the approach they have taken here avoids concerns that might be raised about meta-analysis. Kudos to the authors and hopefully others will follow suite. I just have a few comments/questions that are very minor in nature.

- I’m not as familiar with this literature, but I thought it was a bit strange that the authors referred to their dv (sharing true, but not false news) as “discernment”, which I associate with recognizing the difference between true and false news. It seemed odd to me given that this project came out of the notion that often what people share is not tied to their sense of what is true. That is, for people in the control condition, what they are sharing may not be related to “discernment” at all.

- When the authors looked at Study Level differences, the authors reported simple effects for particular sets of studies. I wondered why the authors didn’t examine/report meta-analytic moderation for these

- For the decay over successive trials section, it would be useful to also report how strong the effect is for the first 4 trials (to make it clear that it is larger, which is the direction I am assuming is correct?)

- I’m not sure whether the format of the journal will allow this or not, but I was wanting a bit more information upfront about 1) how to interpret effect sizes and 2) the design of the studies. Basically, I struggled a little to understand how to interpret the numbers I was looking at when the results came first. I wonder if the authors could include a sentence or two to clarify these things at the top of the results section.

Reviewer #1 (Remarks to the Author):

In this ms., the Authors conduct a meta-analysis of 20 of their own studies bearing on whether accuracy prompts (i.e. reminding people of the importance of sharing accurate information, in different ways) increase sharing discernment (i.e. the sharing of true news by compared to fake news). They find that the effects of accuracy prompts are very robust and generalizable, and derive theoretical implications from their results.

This is a solid ms., and we (this is a joint review) think that the ms. should be accepted down the line. We do have a number of comments.

Thank you!

On the introduction, we believe the Authors do not do a very good job summarizing the existing literature. First, the opening paragraph is misleading. The Authors state, rightly, that fake news have received much attention in the press and the scientific literature. Many readers will likely infer from that that this is because fake news is a significant problem. Instead of implying this conclusion, and talking about the 'meta' level (i.e. about the research, rather than the phenomenon), it would be better if the Authors could succinctly lay out the main conclusions of that research (i.e. regarding the reach of fake news, the likelihood of it affecting elections, etc.).

This is a fair point. Our understanding of the literature is that there is a debate, rather than a set of clear conclusions, about the extent of the misinformation problem. In our revision, we have therefore added language to the intro paragraph explicitly noting that there is debate about the extent of the misinformation problem (but not getting too far into the specifics of different studies, as this debate is not the focus of the current paper).

“There is considerable debate about the scope and impact of the misinformation problem on social media^{2,8-16} (arising in part due to different definitions of “fake news”¹⁷). Be that as it may, a sizable body of research has been devoted to identifying and evaluating approaches for combatting the spread of misinformation online (for reviews, see refs ^{14,18,19}).”

Later on, the Authors are somewhat unfair in their presentation of different theories explaining why people share fake news. When discussing others' theories, they present them as factors that “may also contribute to misinformation sharing,” while “evidence suggests that mere inattention to accuracy [the Authors' theory] plays an important role.” This is not a fair representation of the existing evidence, some of which suggests that political partisanship is the main driver of sharing fake news, while other factors play a more minor role (see for instance the Osmundsen paper cited by the Authors).

We have revised the language to offer a more balanced view of the literature:

“Although factors such as animosity toward political opponents²⁷ and personality factors such as a “need for chaos”²⁸ also contribute to misinformation sharing, evidence suggests that mere inattention to accuracy plays at least some role in the apparent disconnect between accuracy judgments and sharing²⁴.”

In that same paragraph, the Authors write that “purposeful sharing of falsehoods is relatively rare.” It’s not entirely clear what the Authors mean by “purposeful sharing of falsehoods.” For instance, if I share a piece of news even though I’m unsure about its accuracy, because it has other qualities (e.g. being provocative), am I purposefully sharing a falsehood? If I share something from the Onion? If I share something that I caveat? Maybe it would be better to remove that statement, since it’s hard to imagine there’ll be enough space to clarify and defend it here.

We were referring to results in Pennycook et al (2021, Nature) where we find that very few people share news that they indicate is inaccurate. But the referees are right that this may be too general of a claim (and the claim is not important for the point we were making here) so we have removed this sentence in the revision.

Turning to the results, they appear on the whole to be very solid, and some of the figures (e.g. Fig 5) are very informative. A few notes:

A recent paper pointed out some potential limitations of internal meta-analyses

http://urisohn.com/sohn_files/wp/wordpress/wp-content/uploads/ima-internal-meta-analysis-99-published-edited.pdf

The present meta-analysis should be exempt from most of these pitfalls (e.g. all the study were pre-registered) but not from others (e.g. the decision to include the studies in the meta-analysis was likely decided after having run these studies). We don’t believe that the violation of some of these criteria make the present meta-analysis useless, but they should be discussed.

Thank you for directing us to this excellent paper! It would seem that our internal meta-analysis fits the criteria for being reliable based on the arguments of the cited authors. For example, regarding this point: “the decision to include the studies in the meta-analysis was likely decided after having run these studies”. The problem outlined by Vosgerau et al. is when one decides whether or not to run more studies (or to include studies in the analysis) based on how the meta-analysis comes out. For example, they say “internal meta-analysis would be invalid if the decision about which studies to include in the meta-analysis was at all influenced by the studies’ results.” This is not an issue in our case because we determined which studies to include based on a date-range (2017-2020). This date-range was determined based on when we first began running these experiments (2017) and, simply, the end of the year prior to when we decided to complete the meta-analysis (which occurred a few months into 2021). This was decided prior to completing the meta-analysis and was not revised once we looked at the data.

It is noted by Reviewer 2 that there is a wide variety of different types of approaches included here. This is specifically because we did not want to selectively report particular “versions” of the accuracy prompt experiment. Rather, this is simply the entire file-drawer of relevant studies in the U.S. context.

We have added a more extensive discussion of this issue in the introduction:

“To that end, we perform an exhaustive meta-analysis of accuracy prompt experiments that our group has conducted. There are two main threats to the validity of meta-analytic results: the systematic omission of studies (e.g., publication bias suppressing studies with null results^{35,36}), and the flexible selection of analysis approaches within each study inflating the rate of false positives (e.g. p-hacking³⁷). Our meta-analysis addresses both of these issues because we have complete access to all relevant data. This allows us to avoid publication bias by including all qualifying studies, regardless of their results, and avoid inflating false positives through flexible analysis by applying the exact same analytic approach for all studies (an approach that was common across preregistrations for the subset of studies that had pre-registered analysis plans). Although it has been observed that biases caused by flexibility in analyses or selection criteria may be exacerbated in internal meta-analyses³⁸, this is not a concern in the present case. First, there is no bias from analysis flexibility as we use the same analysis as was preregistered in the very first experiment for the full collection of studies. Second, there is no bias from study selection as we determined which studies to include (and when to stop including studies) simply by setting a date range (2017-2020) prior to conducting the meta-analysis. Furthermore, we included all interventions that could be construed as accuracy prompts – i.e. interventions that occurred prior to the sharing task, that invoked accuracy in some way (such that the concept of accuracy would be primed), and did not provide any specific information about the veracity of any particular headlines (were “content neutral”). Importantly, we made the decision about what to include before conducting the meta-analysis. As a result, study selection was broad and not susceptible to motivated or arbitrary choices about inclusion. Our meta-analysis therefore provides an unbiased assessment of the replicability and generalizability of the impact of accuracy prompts on sharing intentions.”

“In each study, only participants who indicated that they use social media were allowed to participate.” Were participants saying that they don’t share news on social media also excluded?

No, participants saying that they don’t share news on social media were *not* also excluded. We have clarified this in the text.

Were most answers recorded on scales, and then recoded as dichotomous outcomes? Could that affect the results in any way?

They were not dichotomized - they were simply rescaled so that min scale value = 0 and max scale value = 1.

Although the existing figures are helpful, it would be nice to have more descriptive data. As it stands, the reader has no idea of what proportion of people share false vs. true news (with or without the intervention). We believe that such descriptive data is really important to get a fuller understanding of a phenomenon.

Great suggestion, we now report the meta-analytic estimates of mean sharing intentions by headline veracity and condition in the main text, and the breakdown by study in SI Table S1.

Regarding the theoretical implications:

The Authors note that “Consider the observation that the treatment effect is smaller for headline sets where baseline discernment is better (and, therefore, that the treatment effect is larger for headline sets where baseline discernment is worse). One possibility that is consistent with the inattention account is that baseline sharing discernment is worse in cases where the content is particularly distracting” First, a possibility is that when baseline discernment is worse there is simply more room for improvement, and thus that this result might be largely artefactual. Second, the finding, if not artefactual, is compatible with other theories, for instance if people where purposefully sharing misinformation.

As shown in the new Table S1, discernment is very poor overall, around 0.055 (average sharing intentions in the control were 0.341 for false and 0.396 for true). Even in the experiment that had highest control discernment, discernment was only 0.11 (false 0.252, true 0.363). Thus, discernment is not close to ceiling and there is substantial room for improvement in every sample.

We are also a bit confused about how this finding can be explained by other accounts. For example, if the sharing of false content was driven by people purposefully sharing misinformation, accuracy prompts should have no effect regardless of baseline discernment.

“Researchers should avoid using MTurk data to make strong claims about differences between Democrats and Republicans.” This is true, but the Authors might not be the first to make that claim—if that’s the case, the Authors’ case would be bolstered by citing previous research.

We have added four relevant citations to this paragraph. Interestingly, the only paper that we could find that specifically investigates the issue (Clifford et al., 2015) found that

MTurk was largely similar to national samples. However, this was only looking at differences relating to personality and values (e.g., moral values).

“This could include investigating what, precisely, causes people to be inattentive to accuracy (e.g., are there characteristics that cause more distraction when reading particular headlines, do things differ from platform to platform, etc.), and what people are attending to instead of accuracy.” Phrasing the question as “what people are attending to instead of accuracy” is misleading. It’s not as if people could only attend to one feature of a piece of information at a time. A better question is: what are the other factors people pay attention to, besides accuracy, when consuming and sharing information. This broad question has been investigated by many disciplines (e.g. use and gratification theory in media studies, cultural attraction theory in cultural evolution, relevance theory in pragmatics, etc.). So the Authors are suggesting here to do something that a great many scholars have been doing for many years.

Good point, we have removed the statement about future work investigating “what people are attending to instead of accuracy”.

typos

“are significant more effective”

“is strong correlated”

“(e.g., it is particularly emotional²⁹, or contains moral content^{30,31}.”

Missing parenthesis

“in worse” should be is worse

Thank you for these notes!

Reviewer #2 (Remarks to the Author):

Review of Nature Communications manuscript NCOMMS-21-40911

Shifting attention to accuracy is a replicable and generalizable approach for reducing the spread of fake news

The manuscript aims to provide a meta-analytic review of evidence behind a range of accuracy prompts. Such prompts can be used to slow the spread of online misinformation without imposing any content restrictions. As such, this is a useful and important contribution to the field, which however at this stage of writing & research suffers from a few issues, which need to be addressed in the revision.

Thank you!

Major points

1. First of all, this is not really a proper meta-analysis of accuracy prompts but of a subset of studies done in one lab. Thus, the study is not following the standard meta-analytic procedure where there is a protocol for inclusion of the studies in the analysis. This might not be a problem per se, since the goal of this particular paper is narrower.

However, I do want to raise a general concern of a potential bias due to the fact that the authors of the meta-analysis and all the studies are the same.

Every meta-analysis makes some choices about which studies to include or exclude in order to derive some estimate of an effect size. Due to the nature of this research, where studies are run with large online samples and the researcher does not directly interact with the participant, it is feasible to conduct a well-powered meta-analysis (in order to derive the estimated effect size) based only on studies completed by one research group. This wouldn't be possible for many meta-analyses, either because the samples would be too limited (or there aren't enough of them) or because it's possible for the researchers to inject some sort of bias into the individual studies by interacting with participants.

So, in essence, the goal is not narrower; i.e., we are setting out to derive some estimate of an effect size based on a collection of studies, as in any other meta-analysis. Rather, it simply achieves that goal in a way that is different than often found in meta-analysis. As noted by Reviewer 3 (the statistical consultant), this is a really robust approach and, in many ways, actually preferential to a meta-analysis of published literature.

In any case, as noted above, we have clarified the logic of our particular approach in the introduction of the manuscript. We have also added a paragraph in the discussion to highlight the limitations of our approach.

We have also added a new analysis in the SI where we have added the one additional experiment conducted during 2017-2020 in the US from outside our lab that we could find (Roozenbeek et al 2021) to the meta-analysis and it does not change the results.

Relatedly, the authors do not engage with research outside of their own lab. For example, pre-registered replication of one of their study by Roozenbeek et al (2021) is not even mentioned in the current manuscript. This is surprising, as I think critical engagement with relevant research on accuracy prompts done outside of the authors' lab is necessary at this stage of evidence evaluation.

This is a good point, thank you. We have added a more extensive discussion of work from other groups that is relevant to our meta-analysis in the introduction:.

“Evidence for the role of inattention comes from experiments in which prompting participants to think about the concept of accuracy – for example, by asking them to evaluate the accuracy of a random headline at the beginning of the study – reduces the disconnect between accuracy judgments and sharing intentions, and thereby increases the quality of news shared²⁴⁻²⁶. This effect has been replicated in preregistered studies conducted by other research groups^{29,30}, a

variety of successful accuracy prompts have been identified²⁵, and the effectiveness of this approach has also been demonstrated in a large field experiment on Twitter where accuracy prompts were sent to users who had been sharing low-quality news content²⁴. However, questions have been raised about whether it operates by decreasing sharing of false news or increasing sharing of true news³⁰, whether it is moderated by individual differences relating to political ideology^{24,29-31} and attentiveness³⁰, and whether it quickly dissipates³⁰.”

Moreover, the authors themselves write “There are two main threats to the validity of meta-analytic results: the systematic omission of studies and the flexible selection of analysis approaches within each study inflating the rate of false positives”. However, it seems that systematically focusing only on their own studies and not even conducting a systematic search for study inclusion, they contradict themselves.

This is not a contradiction: As long as the criteria for inclusion are orthogonal to selection based on results, then the effect size estimate should remain unbiased. The problem with including studies from outside our own lab is that we cannot be sure that there isn't selection based on results (e.g., studies file-drawered because they did not produce significant results). This is consistent with the literature on meta-analysis, as noted by Reviewer 3.

2. Second point concerns conceptual and theoretical underpinning of this research.

Here, a better conceptual explanation as well as concrete description of Accuracy prompts included in the meta-analysis is needed. Table 1 includes different types of interventions subsumed under the “accuracy prompt” umbrella, such as reminders, social norms, nudges and even media literacy tips. They all indeed might have to do with the concept of accuracy but engage participants in fundamentally different ways. I would suggest starting with a definition of what the accuracy prompt means, then outline how it can be enacted in different ways and through different cognitive mechanisms and then how it can be implemented experimentally and in the social media environment. I had to go to the OSF to dig up experimental stimuli to actually see what these different interventions are about – unfortunately, I only found the video and none of the other stimuli. Thus it is really impossible to see what the “Tips” prompt is about (described in the table merely as “Participants are shown a set of minimal digital literacy tips.”) So far, I am not convinced that accuracy prompts and digital literacy tips can be subsumed under the same umbrella – but more information is needed and I think this information should be provided in the paper itself and/or made accessible in the supplement.

We have added a document that provides examples for each intervention in the OSF folder. (Our apologies for the oversight.)

There is, indeed, some variety in approach taken to prompt accuracy. The core feature that is shared across all interventions is that they occur prior to the sharing task, they all

invoke accuracy in some way (such that they prime the concept of accuracy), and they do not provide any specific information about the veracity of any particular headlines (i.e. they are not debunks or corrections) - we now state this in the text.

We see this variety as a core strength of our analysis. It both tests the generalizability of the approach across several different specifications and also ensures that we aren't selecting specific versions of the manipulation for inclusion in the analysis (which, as noted above, could inject bias into the analysis).

At the same time, we do also report analyses that focus specifically on the most common (and, perhaps, prototypical) "Evaluation" manipulation - which clearly only primes accuracy without any other intervention components - and they support the reliability of the manipulation.

I would also like to see at least some explanation of the dependent variable (sharing discernment): why is it of central interest for interventions research (e.g., as opposed to truth discernment)? How is it related to the goal of reducing spread of misinformation and so on.

We have added text to the introduction explaining that we focus on sharing, rather than belief, because simply being exposed to misinformation can increase subsequent belief (e.g. Pennycook et al., 2018). The massive networked character of social media platforms means that when people choose to share misinformation online, it has the potential to reach a large number of others - and as a result, reducing the sharing likelihood of misinformation can substantially reduce it's reach (as show, for example, in the computer simulations of Pennycook et al. 2021 and Bak-Coleman et al. 2021).

3. According to the editor's request, my role was not to evaluate validity and reproducibility of statistical analyses (I am neither an expert in meta-analyses nor do I work with Stata), and I hope another reviewer(s) will do so. Their opinion should also have more weight on this matter. That being said, my impression is that the methodology lacks transparency and could be presented in a more clear and reproducible way. For instance, I would suggest to expand the "Analysis approach" section adding more details on the meta-analytic approach, where the authors could explain their choice of the random effects meta-analysis, discuss heterogeneity of effects, and explain in more detail the summary effects and their main quantity of interest (what they call the meta-analytic estimate). Same goes for all constitutive parts of meta-analytic review process itself (such as inclusion criteria). Here might be the place to discuss your decision not to engage with studies done outside of your lab).

We have added more detail about the random-effects meta-analysis and the meta-analytic estimate to the Analysis Approach section, and have added a detailed discussion of the meta-analytic inclusion criteria etc (including a flow diagram) following the PRISMA guidelines.

In the Results, when reporting main effects (e.g., on p. 5 “We find that accuracy prompts significantly increase sharing discernment (interaction between headline veracity and treatment dummies; Figure 1), $b = 0.038$, $z = 7.102$, $p < .001$, which translates into a 71.7% increase over the meta-analytic estimate of baseline sharing discernment in the control condition, $b = 0.053$, $z = 6.636$, $p < .001$.”), please specify how you have arrived to this number. In general, I think reporting relative numbers (e.g., increase in 72%) tends to inflate the actual effects. Why not report percentage points instead?

Apologies for the confusion, but we are actually already reporting percentage point effects. The coefficient values ($b=...$) that we report are in units of percentage points; the idea of the extra “percent increase” analysis is simply to help contextualize the magnitude of the b 's.

It would also be helpful to report these numbers for control and treatment conditions in the plot (as it is sometimes done in forests plots reporting meta-analytic results), or at least include them in the table format in the SI (along with other relevant statistics, such as heterogeneity of effects in included studies). As a general rule, if a number is reported in the paper it should be supported by a plot or a results table either in the main text or in the supplement.

We now report the meta-analytic estimates of mean sharing intentions by headline veracity and condition in the main text, and the breakdown by study in SI Table S1. We have also added statistics for heterogeneity across studies to the figure legends for each forest plot.

Minor points

In Methods, p. 20, “convenience samples from Amazon Mechanical Turk; samples from Lucid that were quota-matched to the national distribution on age, gender, ethnicity, and region; or representative samples from YouGov” – in what way YouGov samples were nationally representative? Were they also quota-matched?

We have clarified that YouGov uses sample matching for the selection of representative samples from non-randomly selected pools of respondents. Sample matching is a two step process that works as follows. First, YouGov generates a truly random sample of the US population. Then, for each member of the representative sample, YouGov selects a participant from their pool of opt-in respondents who matches the representative sample member on a large set of variables that are available in consumer and voter databases. The resulting sample is therefore more representative than a quota-matched sample, although less representative than a true probability sample.

There are several other minor points, but I do not think it is useful to engage with them until the major points are addressed.

Reviewer #3 (Remarks to the Author):

I was asked in particular to comment on the meta-analytic approach that the authors took in this paper. I love what the authors have done here – this is exactly the kind of approach that psychologists should be taking with their work in general: presenting all of the evidence they have to test a particular effect. And as the authors note, the approach they have taken here avoids concerns that might be raised about meta-analysis. Kudos to the authors and hopefully others will follow suite. I just have a few comments/questions that are very minor in nature.

Thank you!

- I'm not as familiar with this literature, but I thought it was a bit strange that the authors referred to their dv (sharing true, but not false news) as “discernment”, which I associate with recognizing the difference between true and false news. It seemed odd to me given that this project came out of the notion that often what people share is not tied to their sense of what is true. That is, for people in the control condition, what they are sharing may not be related to “discernment” at all.

We have added an explanation of news sharing discernment in the introduction. We mean discernment from an objective, rather than subjective, perspective - hopefully this is now clear in the revised explanation.

“For this analysis, we focus largely on news sharing discernment; i.e., the extent to which the interventions improve the overall quality of news that people share, which is calculated by taking the difference between sharing intentions for true news and false news (with a higher value indicating more relative sharing of true news). This approach is superior to simply focusing on the sharing of false news because an intervention that decreases the sharing of both true and false news equally would not indicate that people are focusing more on accuracy³⁹. Rather, it would indicate that people are simply more skeptical or unwilling to share any news.”

- When the authors looked at Study Level differences, the authors reported simple effects for particular sets of studies. I wondered why the authors didn't examine/report meta-analytic moderation for these

Good point, we have added meta-regressions testing for moderation to the SI (and discussed the results in the appropriate section of the main text).

- For the decay over successive trials section, it would be useful to also report how strong the effect is for the first 4 trials (to make it clear that it is larger, which is the direction I am assuming is correct?)

Good point - we have added this to the revision (and you are correct about the direction).

- I'm not sure whether the format of the journal will allow this or not, but I was wanting a bit more information upfront about 1) how to interpret effect sizes and 2) the design of the studies. Basically, I struggled a little to understand how to interpret the numbers I was looking at when the results came first. I wonder if the authors could include a sentence or two to clarify these things at the top of the results section.

As suggested, we have added a paragraph at the beginning of the results section reminding readers about the basic experimental design, and providing information about how to interpret effect sizes.

REVIEWER COMMENTS

Reviewer #1 (Remarks to the Author):

We're happy with the changes made, and recommend accepting the ms.

Reviewer #2 (Remarks to the Author):

First of all, I would like to thank the authors for their work on revising the manuscript and replying to reviewers' comments. I think the revised manuscript is in a much better shape, especially when it comes to explaining methodology and background research.

I still have a few points I would like to highlight:

1. Reporting results

Papers on such important topics usually attract a lot of attention, also from outside of the research community. Therefore, I believe it is extra important to make sure that the results are expressed in both accessible and precise manner.

Let's take the following sentence as an example (beginning of the Results section):

"We find that accuracy prompts significantly increase sharing discernment (interaction between headline veracity and treatment dummies; Figure 1), $b = 0.038$, $z = 7.102$, $p < .001$, which translates into a 71.7% increase over the meta-analytic estimate of baseline sharing discernment in the control condition, $b = 0.053$, $z = 6.636$, $p < .001$."

It is not very clear what's going on here and I am still confused about the second set of values (what you call "the meta-analytic estimate of baseline sharing discernment in the control condition"). In the "Response to reviewers", you write "We now report the meta-analytic estimates of mean sharing intentions by headline veracity and condition in the main text, and the breakdown by study in SI Table S1." So I assume that the baseline numbers come from the S1 Table. In the table, I can see the grand mean (what you call "Meta" – please find a more precise term for this, e.g., meta-analytic mean and call it this way in the text too to avoid confusion). However, I can't be sure, as the difference between sharing intentions for false and true headlines is 0.055 (and not 0.053). Moreover, in the quoted fragment you express it in the same way as the meta-analytic estimate from Figure 1 ($b=$). So I am still confused about this number. Might be helpful to report it in the table in the exactly same way as in the text.

Once you clarify it, I also suggest putting the relative increase into context, for instance:

Accuracy prompts significantly increase sharing discernment (interaction between headline veracity and treatment dummies; Figure 1) by 0.038 points on a 0-1 scale ($z = 7.102$, $p < .001$), which translates into the absolute increase in the meta-analytic mean sharing discernment from baseline value of 0.053 to 0.091 (see Table S1) or a relative increase in the magnitude of 71.7%.

The same goes for all results reporting, as well as the abstract (reporting 72% number alone can be misleading).

It would be great if you could additionally provide a wider context for this effect (maybe in the Discussion), similar to The Number Needed to Treat in medical interventions research. For instance, how

many people online need to be exposed to this intervention to decrease misinformation by x %.

In some cases, you do not mention the values for control condition (e.g., p.9, 10 etc). I would suggest still doing it every time and reference the Table/Figure where they are reported (and always putting relative % into context, like in my suggestion above).

2. Table S1 is generally very helpful, both in terms of descriptive stats and the control estimates. I suggest including it in the main text. One could even think about adding a graph to it with meta-analytic means.

3. Table 2 – I would suggest adding a new column, e.g., “Pre-registered?” with YES/NO and a link

4. Discussion: I appreciate discussion on limitations of your approach, but I would also like to see some discussion related to the generally small effect sizes in your studies. Otherwise, one (imagine, a policy maker) can leave with an impression that accuracy prompts are a silver bullet. However, effect on sharing intentions for false news is really very small. I would therefore suggest being upfront about it and maybe highlight advantages of combining different interventions.

When discussing limitations of internal meta-analysis, please add the fact that your meta-analysis was not pre-registered for full transparency. I do think this is somewhat problematic, as also highlighted by Vosgerau et al, 2019.

About the replication study you’ve included:

“While the one replication of an accuracy prompt intervention we are aware of that meets our inclusion criteria (conducted in the U.S. between 2017 and 2020) did find a significant positive effect on sharing discernment – and including this study in our main analysis does not meaningfully alter the results (see SI Section 4) – it would be valuable for future work to examine studies conducted by a wider range of research groups.”

First, minor thing: please add a citation to the mentioned study here. Second, please specify which sample you included in your analyses (I assume, Stage 2, pooled?).

Minor comments

In Table 1, define Literacy Tips more precisely, e.g., by simply quoting the intervention text.

Figure 3, caption: should be true instead of false news (same in the SI)

Figure 5 – consider changing one of the colors so that the figure also works in case somebody prints it in black and white.

Too many “finally, we ask” – followed by one more “finally”. I think there should be just one.

Reviewer #3 (Remarks to the Author):

The authors have done a great job addressing the few minor comments I had - thanks!

First of all, I would like to thank the authors for their work on revising the manuscript and replying to reviewers' comments. I think the revised manuscript is in a much better shape, especially when it comes to explaining methodology and background research.

I still have a few points I would like to highlight:

We appreciate the helpful comments for how to make the manuscript clearer.

1. Reporting results

Papers on such important topics usually attract a lot of attention, also from outside of the research community. Therefore, I believe it is extra important to make sure that the results are expressed in both accessible and precise manner.

Let's take the following sentence as an example (beginning of the Results section):

"We find that accuracy prompts significantly increase sharing discernment (interaction between headline veracity and treatment dummies; Figure 1), $b = 0.038$, $z = 7.102$, $p < .001$, which translates into a 71.7% increase over the meta-analytic estimate of baseline sharing discernment in the control condition, $b = 0.053$, $z = 6.636$, $p < .001$."

It is not very clear what's going on here and I am still confused about the second set of values (what you call "the meta-analytic estimate of baseline sharing discernment in the control condition").

The numbers we report are the meta-analytic estimates of coefficients on various terms in the regression models we describe in the Methods section:

For analysis purposes, sharing decisions (the dependent variable) are rescaled such that the minimum possible value is 0 (indicating a very low likelihood of sharing) and the maximum possible value is 1 (indicating a very high likelihood of sharing). Within each study, we conduct a rating-level (i.e. one observation per subject-item pair) linear regression with robust standard errors clustered on participant and headline, taking sharing intention as the dependent variable. Our main analysis includes a dummy for headline veracity (0=false, 1=true), a dummy for condition (0=control, 1=accuracy prompt), and the interaction term. With this specification, the coefficient on the interaction term indicates the accuracy prompt's effect on sharing discernment (the difference in sharing likelihood for true relative to false headlines), the coefficient on the condition dummy indicates the accuracy prompt's effect on sharing intentions for false headlines, and the coefficient on the headline veracity dummy indicates baseline sharing discernment in the control condition. Our participant-level heterogeneity analyses use models that add the individual difference being interrogated along with all interaction terms, and focus on the 3-way interaction (the extent to which the individual difference moderates the accuracy prompt's effect on sharing discernment); and our analysis of order effects adds trial number along with all interaction terms, and again focuses on the 3-way interaction.

For any given coefficient of interest, we calculate an estimate for each study. Our interest, however, is not the effect in any given study. Instead, we are interested in the best estimate of the effect using the data from all studies. Therefore, for each coefficient of interest, we combine the estimates from each study using random-effects meta-analysis to generate this overall estimate – what we refer to as the "meta-analytic estimate" of the value of that

coefficient. We use random-effects meta-analysis, rather than fixed-effects meta-analysis, because there is reason to expect that the true effect size varies across studies (because, for example, different studies used different versions of the treatment, different headlines, and different subject pools).

To help clarify this, in the Results we now indicate which specific coefficient we are reporting for each test (underlines added here for emphasis and not included in the actual text), e.g.

We find that accuracy prompts significantly increase sharing discernment (interaction between headline veracity and treatment dummies: $b = 0.038$, $z = 7.102$, $p < .001$; Figure 1), which translates into a 71.7% increase over the meta-analytic estimate of baseline sharing discernment in the control condition (headline veracity dummy: $b = 0.053$, $z = 6.636$, $p < .001$). This increase in discernment was driven by accuracy prompts significantly decreasing sharing intentions for false news (treatment dummy: $b = -0.034$, $z = 7.851$, $p < .001$; Figure 2), which translates into a 10% decrease relative to the meta-analytic estimate of baseline sharing intentions for false news in the control condition (intercept: $b = 0.341$, $z = 15.695$, $p < .001$). Conversely, there was no significant effect on sharing intentions for true news (treatment dummy from model with “true” as the holdout category for headline veracity: $b = 0.006$, $z = 1.44$, $p = .150$; Figure 3).

In the “Response to reviewers”, you write “We now report the meta-analytic estimates of mean sharing intentions by headline veracity and condition in the main text, and the breakdown by study in SI Table S1.” So I assume that the baseline numbers come from the S1 Table. In the table, I can see the grand mean (what you call “Meta” – please find a more precise term for this, e.g., meta-analytic mean and call it this way in the text too to avoid confusion). However, I can’t be sure, as the difference between sharing intentions for false and true headlines is 0.055 (and not 0.053). Moreover, in the quoted fragment you express it in the same way as the meta-analytic estimate from Figure 1 ($b=$). So I am still confused about this number. Might be helpful to report it in the table in the exactly same way as in the text.

Because (i) the meta-analysis used each study’s standard error when assigning weights to each study, and (ii) the standard errors are different for different terms from a given study, the weighting of studies is not identical across meta-analyses for different regression coefficients. That is why, for example, the meta-analytic estimate for baseline sharing discernment (i.e. the coefficient on the headline veracity dummy from the main regression model) is slightly different from the difference in the grand means for sharing intentions of true versus false headlines.

Once you clarify it, I also suggest putting the relative increase into context, for instance: Accuracy prompts significantly increase sharing discernment (interaction between headline veracity and treatment dummies; Figure 1) by 0.038 points on a 0-1 scale ($z = 7.102$, $p < .001$), which translates into the absolute increase in the meta-analytic mean sharing discernment from baseline value of 0.053 to 0.091 (see Table S1) or a relative increase in the magnitude of 71.7%.

To provide this context, we report

Average baseline sharing intentions was 0.341 for false headlines and 0.396 for true headlines (meta-analytic discernment estimate = 0.053); average sharing intentions following an accuracy prompt was 0.309 for false headlines and 0.404 for true headlines (meta-analytic discernment estimate = 0.091).

The same goes for all results reporting, as well as the abstract (reporting 72% number alone can be misleading).

Given the space constraints of the abstract, we don't see how to do this in a clear way. To help contextualize the discernment effect size in the abstract, we have also added the percent decrease in sharing of false statements

Overall, accuracy prompts increased the quality of news that people share (sharing discernment) by 72% relative to control, primarily by reducing sharing intentions for false headlines by 10% relative to control.

We believe that in this case, the percent decrease number is actually more relevant/interpretable than the raw number of unit change on the [0-1] scale. For example, if baseline sharing of falsehoods was 0.03 on the 0-1 scale, a decrease in sharing of 0.03 would mean total elimination of falsehoods; whereas baseline sharing of falsehoods was .9 on the 0-1 scale, a decrease of 0.03 would make comparatively little difference.

It would be great if you could additionally provide a wider context for this effect (maybe in the Discussion), similar to The Number Needed to Treat in medical interventions research. For instance, how many people online need to be exposed to this intervention to decrease misinformation by x %.

Although we appreciate the general idea, this turns out to be very complicated to calculate because of the networked nature of social media/misinformation spread, such that the impact of any given person's sharing depends on what the network structure is, where they are in the network, how many followers they have, etc. We did various agent based simulations in the SI of our 2021 Nature paper to give insight into these kinds of questions, and we have added a reference to this in the Discussion (when discussing effect sizes etc, as per below).

In some cases, you do not mention the values for control condition (e.g., p.9, 10 etc). I would suggest still doing it every time and reference the Table/Figure where they are reported (and always putting relative % into context, like in my suggestion above).

We have added stats for the control condition, as per our result above, throughout.

2. Table S1 is generally very helpful, both in terms of descriptive stats and the control estimates. I suggest including it in the main text. One could even think about adding a graph to it with meta-analytic means.

Given that there is already a lot of figures/tables in the main text, and that we present the key meta-analytic summary stats from the Table S1 in the main text, we have opted to

keep the Table in the SI.

3. Table 2 – I would suggest adding a new column, e.g., “Pre-registered?” with YES/NO and a link

We have decided against this because the table is already quite wide, and whether or not the individual studies were preregistered is not relevant for our meta-analysis since we are applying the same analytic strategy to every study (as opposed to meta-analyzing across studies with variations in analytic strategy, in which case knowing if the individual analytic strategies were preregistered is a relevant evaluation criterion).

4. Discussion: I appreciate discussion on limitations of your approach, but I would also like to see some discussion related to the generally small effect sizes in your studies. Otherwise, one (imagine, a policy maker) can leave with an impression that accuracy prompts are a silver bullet. However, effect on sharing intentions for false news is really very small. I would therefore suggest being upfront about it and maybe highlight advantages of combining different interventions.

We have added the following to the Discussion:

Of course, no single approach will solve the misinformation problem. Accuracy prompts should be therefore considered in combination with a wide range of other approaches^{14,44}. Moreover, the effects we document here, while being replicable and generalizable, are modest in size (although it is unclear how the magnitude of effects observed in the survey experiments we conducted here relate to the actual effect sizes that would be observed on platform, especially given the possibility of network effects that amplify individual-level effects; see ref 24 SI Section 6 for illustrative network simulations). If technology companies explore accuracy prompt interventions, they should conduct experiments to optimize the treatment format and delivery, with the goal of maximizing treatment effect sizes and durability.

When discussing limitations of internal meta-analysis, please add the fact that your meta-analysis was not pre-registered for full transparency. I do think this is somewhat problematic, as also highlighted by Vosgerau et al, 2019.

We now note that the meta-analysis was not pre-registered.

About the replication study you've included:

“While the one replication of an accuracy prompt intervention we are aware of that meets our inclusion criteria (conducted in the U.S. between 2017 and 2020) did find a significant positive effect on sharing discernment – and including this study in our main analysis does not meaningfully alter the results (see SI Section 4) – it would be valuable for future work to examine studies conducted by a wider range of research groups.”

First, minor thing: please add a citation to the mentioned study here. Second, please specify which sample you included in your analyses (I assume, Stage 2, pooled?).

Citation and sample details (indeed we use the pooled data) added.

Minor comments

In Table 1, define Literacy Tips more precisely, e.g., by simply quoting the intervention text.

The text is too long to fit in the table unfortunately – we now refer readers to Epstein et al. 2021 which shows the text of the tips.

Figure 3, caption: should be true instead of false news (same in the SI)

Fixed

Figure 5 – consider changing one of the colors so that the figure also works in case somebody prints it in black and white.

We printed out the figure and it seems to render fine.

Too many “finally, we ask” – followed by one more “finally”. I think there should be just one.

Replaced all but one “finally”

REVIEWERS' COMMENTS

Reviewer #2 (Remarks to the Author):

I am satisfied with the authors response and have no further comments.